# Biomimetic selenocystine based dynamic combinatorial chemistry for thiol-disulfide exchange

Andrea Canal-Martín [1] & Ruth Pérez-Fernández [1✉]

Dynamic combinatorial chemistry applied to biological environments requires the exchange chemistry of choice to take place under physiological conditions. Thiol-disulfide exchange, one of the most popular dynamic combinatorial chemistries, usually needs long equilibration times to reach the required equilibrium composition. Here we report selenocystine as a catalyst mimicking Nature's strategy to accelerate thiol-disulfide exchange at physiological pH and low temperatures. Selenocystine is able to accelerate slow thiol-disulfide systems and to promote the correct folding of an scrambled RNase A enzyme, thus broadening the practical range of pH conditions for oxidative folding. Additionally, dynamic combinatorial chemistry target-driven self-assembly processes are tested using spermine, spermidine and NADPH (casting) and glucose oxidase (molding). A non-competitive inhibitor is identified in the glucose oxidase directed dynamic combinatorial library.

---

[1] Structural and Chemical Biology Department, Centro de Investigaciones Biológicas "Margarita Salas", CIB-CSIC, Madrid 28040, Spain. ✉email: ruth.perez@csic.es

Dynamic combinatorial chemistry (DCC) establishes molecular networks under thermodynamic control that responds to external stimuli[1–3]. We are interested in the application of DCC systems or Dynamic combinatorial libraries (DCLs) to biological environments where a protein or a biomolecule directs the assembly of the building blocks at dynamic equilibrium towards the synthesis of the best ligand or synthetic receptor in situ. Protein-directed DCC has proven its effectiveness as a hit identification strategy discovering enzyme inhibitors[4–7].

Disulfide exchange[8–10] is considered one of the most popular dynamic covalent chemistries applied to biological systems. This dynamic process is based on the thiol–disulfide equilibrium where the slow oxidation of the thiols competes with the disulfide exchange in aqueous solutions. Thiol/disulfide exchange is favored by highly nucleophilic thiolates attacking disulfide bonds constituted by good electrophiles and stable sulfur leaving groups. Therefore, to allow the mixture to self-correct for reaching the equilibrium composition, disulfide exchange must be faster than the oxidation of the thiols to avoid a kinetic trap[11]. Even though disulfide exchange proceeds smoothly at neutral or slightly basic pH, it usually requires several days to reach the required equilibrium[12,13]. To improve this limitation, several reaction conditions based on the use of a co-solvent such as DMSO[14], glutathione redox buffer[15], and high concentrations of selenol derivatives[16,17] have been reported as alternative additives to speed up the exchange reaction from weeks to days. However, the application of certain additives in biological environments is limited. The addition of a suitable catalyst[18] or additive was reported for acylhydrazone exchange to enable the exchange at neutral pH in 5 h even at low temperatures[19]. The possibility of conducting the thiol–disulfide exchange under a reasonable time-frame would benefit from its application in biotemplated-driven dynamic chemical systems and systems chemistry research[20–24].

Inspired by Nature's strategy to accelerate a thiol–disulfide exchange, we focus our attention on one of the major antioxidant systems in mammalian cells, the thioredoxin system which is formed by the mammalian thioredoxin reductase (TrxR), thioredoxin (Trx), and NADPH (nicotinamide adenine dinucleotide phosphate)[25]. Mammalian thioredoxin reductase (TrxR) is a pyridine nucleotide disulfide oxidoreductase that uses selenocysteine (Sec)[26] in place of cysteine (Cys) in catalysis of the reduction of its target protein, thioredoxin (Trx) (Fig. 1)[27,28]. Sec can accelerate the rate of the thiol–disulfide exchange reactions at different stages of the TrxR mechanism. Selenium acts as a nucleophile initially attacking the disulfide bond of Trx (Fig. 1a), or as an electrophile accepting electrons from the redox center of TrxR as part of the selenosulfide bond (Fig. 1b).

The pH range influences the relative difference in reactivity between Cys and Sec in proteins. The $pK_a$ differences between Sec $pK_a \sim 5.43$[29], Cys $pK_a \sim 8.22$[30] contributes to the fact that at certain pHs the concentration of thiolates is low compared to the same reaction containing selenium. The higher polarizability of selenoates made them better nucleophiles and leaving groups than the corresponding sulfur derivatives. Additionally, the bond dissociation energy of Se–Se (46 kcal mol$^{-1}$) compared to S–S (64 kcal mol$^{-1}$) is considerably small[31]. Hence, the thiol oxidoreductase-like catalysis profits from the use of Sec whether it is due to the nucleophilicity, electrophilicity, $pK_a$, or selenium leaving group ability. The exchange reactions of thiol–disulfide and selenol–diselenide share the nucleophilic addition mechanism (Fig. 2). Diselenide exchange has been reported as a DCC reversible reaction in water[32]. A comparative NMR kinetic study of thiol/disulfide and selenol/diselenide reactions showed that selenol–diselenide exchange reactions were ~10$^7$ fold faster than the thiol/disulfide exchange reactions (Fig. 2)[33].

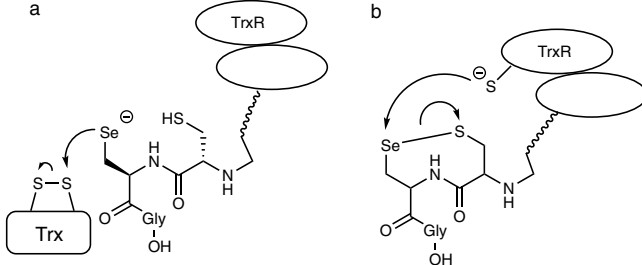

**Fig. 1 Schematic role of selenium in the TrxR-proposed mechanisms. a** Selenium acting as a nucleophile. **b** Selenium acting as an electrophile.

**Fig. 2 DCC schemes. a** Scheme of disulfide exchange. **b** Diselenide exchange. **c** Selenenylsulfide exchange in a thiol–disulfide DCC system.

In this work, we report the use of selenocystine (Sec$_{ox}$) as a promoter of thiol–disulfide exchange at low temperatures and basic pH. Selenium generates selenenylsulfide intermediates which are more reactive towards thiolate nucleophilic addition accelerating the exchange between the different species. This chemistry is inspired by the operating mechanism proposed for the thioredoxin family of proteins. The catalytic efficiency of selenocystine is studied and applied to different thiol–disulfide DCLs including a slow DCC system constituted by alkyl thiols. Furthermore, we evaluate selenocystine as a promoter for the right formation of disulfide bonds during the folding of a scrambled RNase A even at acidic pH. DCC target-driven self-assembly (casting and molding) processes in selenocystine's presence are tested. As a proof of concept, the casting approach where library members assemble around a template is tested using spermine, among other biomolecules (e.g., spermidine, NADPH). On the other hand, the molding approach in which the assembly of library members occurs inside the binding pockets of the template is studied with glucose oxidase (GOx) from *Aspergillus niger*. The affinity of the amplified molecule (**4**)$_2$ is measured using fluorescence techniques and the glucose oxidase activity is evaluated in tandem with the horseradish-peroxidase system confirming the non-competitive inhibition of (**4**)$_2$.

## Results

Selenocystine based DCLs were prepared to mimic the mammalian thioredoxin reductase (TrxR) system which uses selenocysteine instead of cysteine in its catalysis of the reduction of Trx. DCLs were composed of dithiols (**1–2**)[34–36] and monothiols (**3–4**) to form different architectures such as cyclic and linear oligomers, expanding the possibilities for the molecular recognition of different templates (Fig. 3b). Besides, the addition of carboxylate groups contributed to their solubility under the DCL conditions and the interaction with positively charged amines from the templates chosen through hydrogen bonding and ionic interactions. The aromatic rings may participate in hydrophobic interactions with the corresponding templates. Therefore, DCLs using thiols **1–4** were prepared in the absence and presence of

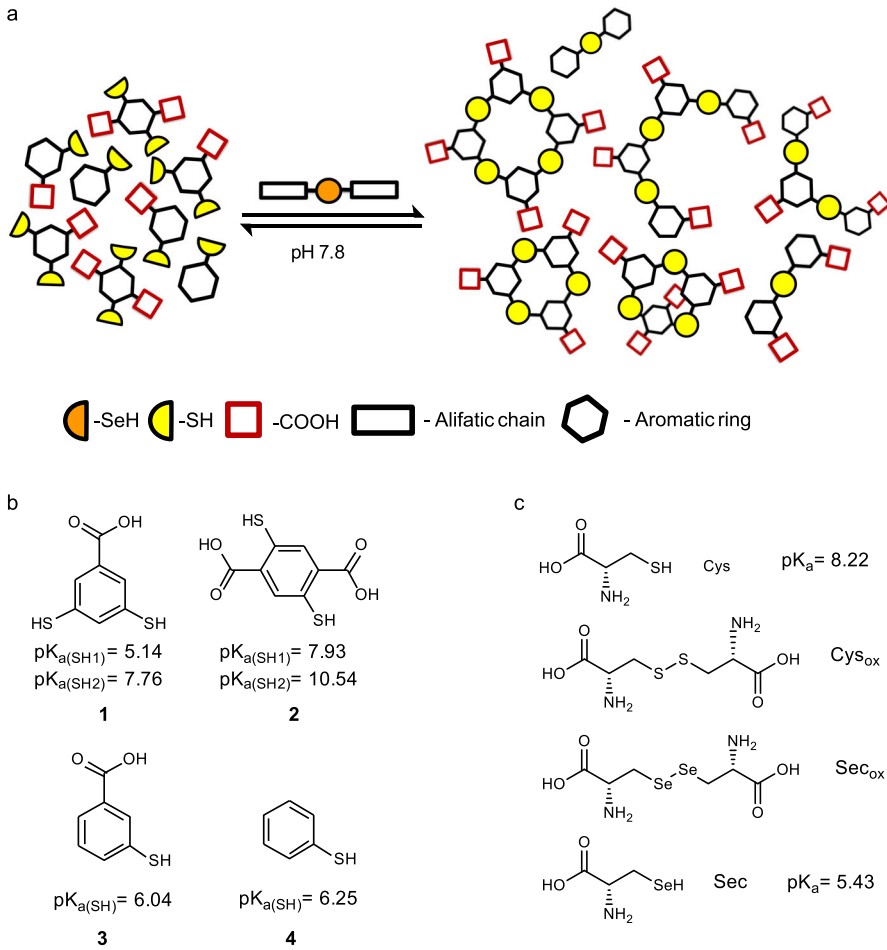

**Fig. 3 Thiol/disulfide catalyzed DCC. a** DCC general scheme. **b** Building blocks of DCL. **c** Sulfur and selenium Cys and Sec derivatives. The $pK_a$ corresponds to the thiol group. Estimated $pK_a$s data calculated with Epik as implemented in Schrödinger Suite Release 2020-2[54-56] and reported in the literature (Sec $pK_a$[29], Cys $pK_a$[30]). See Supplementary Fig. 1 for $pK_a$ calculations.

cysteine and selenocystine derivatives (Fig. 3c). Their efficiency speeding up the thiol/disulfide exchange in a rich mixture of oligomers was tested at low temperatures and no stirring to maintain the stability of potential biological templates.

**Sec$_{ox}$ as a catalyst**. We started our DCL by reacting building blocks **1–4** (Fig. 3b) in the absence and the presence of the oxidized and reduced forms of sulfur and selenium derivatives of cysteine (Cys), cystine (Cys$_{ox}$), and selenocystine (Sec$_{ox}$) at basic pH and 6 °C. Sec was not used in the DCL since selenols are easily oxidized and thus more difficult to manipulate[16].

The DCLs were set up at two different pHs 7.8 and 8.8 to ensure Cys deprotonation. Selenium and sulfur derivatives, Sec$_{ox}$ and Cys$_{ox}$ (Fig. 3c) were added as the fifth building block in each of the experiments. Differences in equilibration time at pH 7.8 or 8.8 in tris buffer were negligible (Supplementary Fig. 3 and Supplementary Methods). Therefore, we set up the experiments at pH 7.8 and use diselenide Sec$_{ox}$ and disulfide Cys$_{ox}$ for comparison (Fig. 4).

The DCL that introduced Sec$_{ox}$ as a building block reached equilibrium after 24 h (Fig. 4c). Conversely, DCLs with its homolog Cys$_{ox}$ needed 3 days to equilibrate (Fig. 4b) similar to the control library which required 4 days (Fig. 4a). Different oligomers were identified by HPLC-MS in the DCL (Supplementary Figs. 4–20 and Supplementary Discussion). Studies were performed to ensure the reversibility of the disulfide exchange in the DCLs. In the presence of Sec$_{ox}$, consecutive DCLs were

generated where an additional thiol was added each time the DCL has reached its equilibration point. The DCL started with thiol **1**, followed by the addition after equilibration of thiols **3** and **4**. Finally, the addition after equilibration of thiol **2** showed an identical distribution to Fig. 4 (Supplementary Fig. 35 and Supplementary Methods).

**Optimal concentration of Sec$_{ox}$ in the DCL**. The minimum concentration of Sec$_{ox}$ necessary for the exchange to occur in 24 h was studied. Several concentrations of Sec$_{ox}$ from 1 to 10% (mol) according to the total concentration of thiols in the DCL were evaluated. In the absence of selenocystine, the system took 96 h to reach equilibrium (Fig. 5a, (i)). As the concentration of Sec$_{ox}$ increased, the equilibration time was reduced from 96 to 72 h (1% mol), 48 h (2.5% mol), and 24 h (5 and 10% mol). The results showed that a concentration of 5% mol of Sec$_{ox}$ was required for the system to reach equilibrium within 24 h (Fig. 5a, (iv)). RPAs were calculated to confirm the library equilibration and to verify that the proportion of the species generated did not depend on the different Sec$_{ox}$ percentages. Selenyl-sulfide intermediates were detected through LC–MS supporting the existence of these reactive intermediates to promote the thiol–disulfide exchange (Supplementary Figs. 31–34).

**Kinetic rate of Sec$_{ox}$**. To quantify the catalytic efficiency of Sec$_{ox}$ in thiol–disulfide exchange, kinetic studies were performed under our standard experimental conditions. The homodimerization of

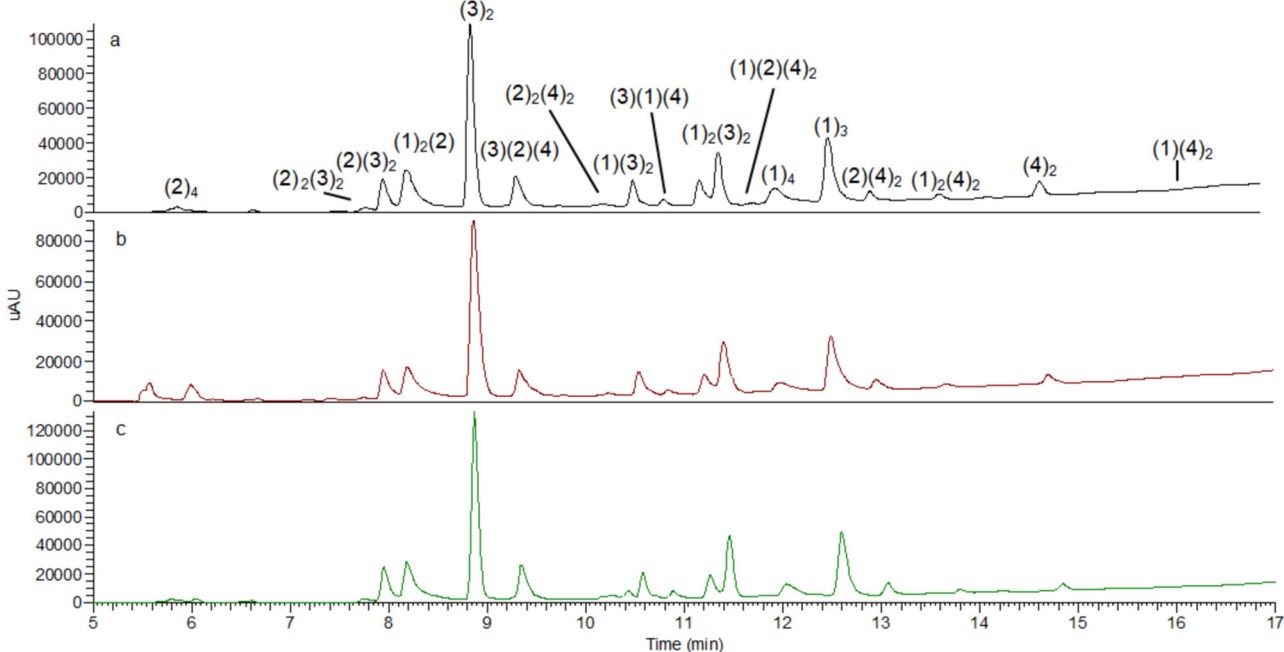

**Fig. 4 DCL chromatograms comparing disulfide (Cys$_{ox}$) and diselenide (Sec$_{ox}$).** DCL conditions: building blocks (**1–2**) 95 μM each (**3–4**) at 190 μM concentration each. Tris buffer 20 mM, pH 7.8, 6 °C. **a** Control after 96 h. **b** Cys$_{ox}$ (190 μM) after 72 h. **c** Sec$_{ox}$ (190 μM) after 24 h. Experiments were performed in triplicate.

building block **3** to (**3**)$_2$ (Fig. 6a) was followed by HPLC-MS in the absence and presence of the previously mentioned percentages of Sec$_{ox}$.

The accelerating role of DMSO[14] was discarded by running control experiments containing up to 20% (v/v) DMSO, which is much larger than the amount of this co-solvent used in our standard conditions (2.5% (v/v)). Furthermore, we have compared the catalytic activity of selenocystine to that of selenoenzyme TrxR.

Figure 6b shows the time course for the (**3**)$_2$ homodimerization during the first 13 h and the derived kinetic parameters (Fig. 6c). In the presence of 5% (mol) Sec$_{ox}$, the homodimerization reaction is completed in 24 h. The addition of 5% Sec$_{ox}$ ($K_{obs}$: 0.259 ± 0.009 M$^{-1}$ s$^{-1}$) led to a 18-fold acceleration over the uncatalyzed reaction ($K_{obs}$: 0.014 ± 0.001 M$^{-1}$ s$^{-1}$). The reaction rate obtained with 5% (mol) of selenocystine is superior to those observed with 20% (v/v) DMSO ($K_{obs}$: 0.068 ± 0.001 M$^{-1}$ s$^{-1}$) and 5% (mol) recombinant human TrxR 2 ($K_{obs}$: 0.035 ± 0.003 M$^{-1}$ s$^{-1}$). Of note, the kinetic rate observed with 5% (mol) selenoenzyme resulted lower than with 1% (mol) Sec$_{ox}$ ($K_{obs}$: 0.091 ± 0.002 M$^{-1}$ s$^{-1}$).

**Sec$_{ox}$ effect in a slow DCL**. To broaden the scope of selenocystine we tested this catalyst in a slow disulfide-exchange system. The DCL comprised 4 building blocks (**5–8**)[37], three of them being aliphatic thiols with p$K_a$ values around 9 to slow down the exchange reaction (Fig. 7a). Due to solubility issues, a 20% (v/v) DMSO was added and the reaction mixture was diluted. In the absence of 5% (mol) Sec$_{ox}$ (with respect to total thiol concentration), the system reached equilibrium after 192 h, whereas in the presence of selenocystine the system equilibrated after 60 h, i.e., three times faster. The different oligomers were identified by HPLC-MS (Supplementary Figs. 21–30).

**Folding of scrambled RNase A with Sec$_{ox}$**. The folding of a small protein such as RNase A was studied to explore Sec$_{ox}$ application in the formation of the correct disulfide bonds during protein folding. The scrambled RNase A, a mixture of oxidized forms of RNase A with a random distribution of four disulfide bonds[38], was used to determine the refolding rate. Once the optimum concentration of GSH and Sec$_{ox}$ for protein folding was determined (Supplementary Fig. 40), the relative folding rates for (Sec$_{ox}$/GSH) pair were measured and compared to the standard glutathione redox buffer (GSSG/GSH) pair (Fig. 7b)[39,40]. Aliquots were withdrawn from each of the experiments and immediately assayed for RNase A activity by following the hydrolysis of cyclic cytidine-2′,3′-monophosphate (cCMP) at pH 6 for 2 min in a spectrophotometric discontinuous assay[41,42].

RNase A folding versus time was plotted where the percent activity was proportional to the activity of folded native RNase A (Fig. 7b). Improved refolding rates were observed replacing GSSG with Sec$_{ox}$. Sec$_{ox}$ provided a 92% yield of the correctly folded enzyme during the first 65 min whereas GSSG/GSH reached the same yield (93%) in 341 min.

To confirm the benefit of using Sec$_{ox}$ over GSSG the same assays were performed at pH 5.4. Whereas GSSG/GSH only got half of the protein refolded, Sec$_{ox}$ reached a 75% yield of native RNase A in <2 h (Supplementary Fig. 41 and Supplementary Methods).

**Sec$_{ox}$ in spermine-directed DCC system**. To test the casting DCC target-driven self-assembly, the template spermine was introduced as a guest into a DCL[36,43–45]. As spermidine and its precursor, the putrescine, these polyamines play important roles in many cellular processes such as the regulation of the kinases' activities, the protection from oxidative damage, the regulation of transcription and translation contributing to nucleic acid stability, modulation of ion channels activity and the preservation of membrane structure/function.

We chose the spermine two building blocks DCL reported by Vial et al.[36] as a well-studied system to test that Sec$_{ox}$ did not interfere with the recognition process and that the same DCL with two building blocks was efficiently equilibrated in the presence of 5% (mol) selenocystine. Building blocks **2** and **3**

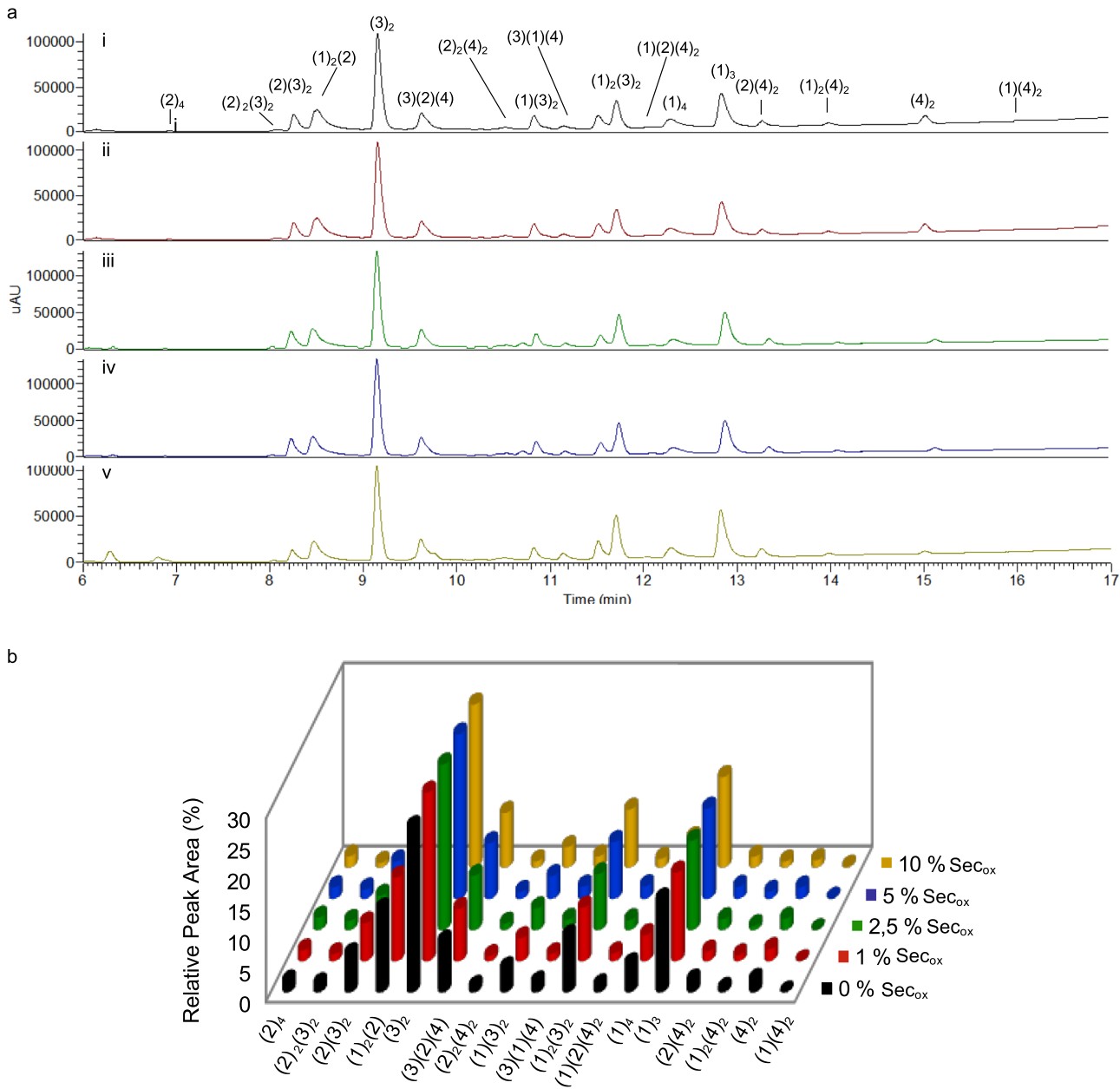

**Fig. 5 Analysis of Sec$_{ox}$ percentages in a 4 building blocks DCL. a** DCLs chromatograms showing different percentages of Sec$_{ox}$. DCL conditions: building blocks (1–2) at 95 μM concentration each and building blocks (3–4) at 190 μM concentration each. Tris buffer 20 mM pH 7.8, 6 °C, (i) 0% mol as control DCL, 96 h (black), (ii) 1% mol (5.7 μM) (red) in 72 h, (iii) 2.5% mol (14.25 μM) (green) in 48 h, (iv) 5% mol (28.5 μM) (blue) in 24 h, (v) 10% mol (57 μM) (yellow) in 24 h. **b** Relative peak areas (RPAs) of the DCLs after equilibration in the absence and in the presence of different Sec$_{ox}$ percentages (Supplementary Table 1). Experiments were performed in triplicate.

equipped with carboxylate moieties to establish interactions with the protonated amine groups of the spermine were allowed to equilibrate at pH 7.8 and 6 °C (Fig. 8a). As expected in our control experiment, the major component in the absence of spermine was the linear tetramer $(2)_2(3)_2$. After the addition of spermine, the linear tetramer "decomposed" in order to form mainly the cyclic tetramer of $(2)_4$, and the $(3)_2$ dimer (Fig. 8a). As it was reported, several studies determined that the spermine was threaded through the macrocycle fixing the configuration to a highly symmetric stereoisomer of $(2)_4$[36]. The addition of 5% (mol) Sec$_{ox}$ not only speeded four times the DCL equilibration (from 96 to 24 h) but also did not interfere with the recognition event reaching the same result as our control experiment (absence of Sec$_{ox}$), matching the previously reported data and amplifying

mainly the cyclic tetramer $(2)_4$ (Fig. 8a). The precise composition of the DCL (with and without spermine), was assessed by measuring the relative peak area (RPA). Indeed, the normalized change of RPA was used to quantify the spermine influence in the outcome (Supplementary Fig. 42)[46]. The same template (spermine) was tested in our reference DCL of four building blocks (1–4) in the absence and presence of Sec$_{ox}$ (Fig. 8b). The addition of selenocystine speeded the equilibration time of the DCL to 24 h and the normalized change of RPA showed the amplification of the cyclic tetramer $(2)_4$ (Supplementary Fig. 43). Additionally, two different templates were tested to confirm the absence of conflict between selenocystine and the templating effect. Spermidine and NADPH were introduced in our 4 building blocks DCL (1–4) in the absence and presence of Sec$_{ox}$, leading to

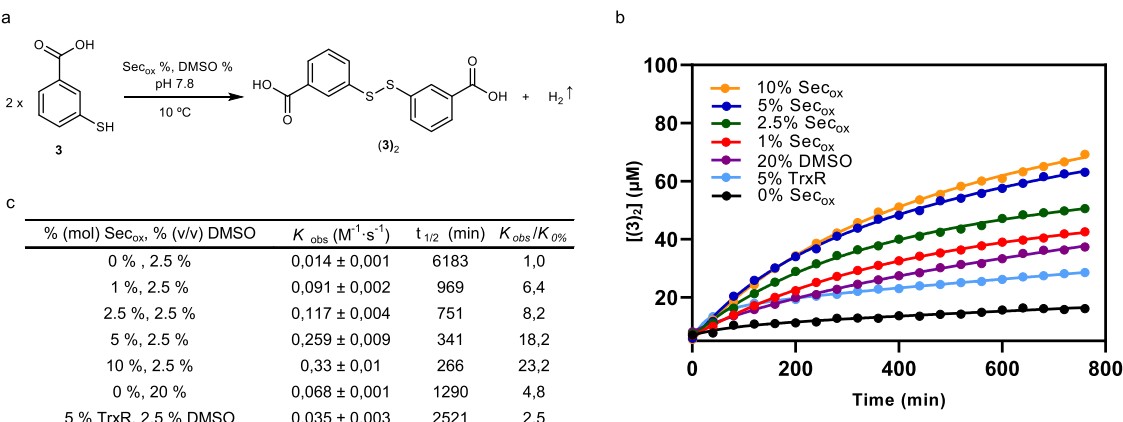

| % (mol) Sec$_{ox}$, % (v/v) DMSO | $K_{obs}$ (M$^{-1}$·s$^{-1}$) | $t_{1/2}$ (min) | $K_{obs}/K_{0\%}$ |
|---|---|---|---|
| 0 % , 2.5 % | 0,014 ± 0,001 | 6183 | 1,0 |
| 1 %, 2.5 % | 0,091 ± 0,002 | 969 | 6,4 |
| 2.5 %, 2.5 % | 0,117 ± 0,004 | 751 | 8,2 |
| 5 %, 2.5 % | 0,259 ± 0,009 | 341 | 18,2 |
| 10 %, 2.5 % | 0,33 ± 0,01 | 266 | 23,2 |
| 0 %, 20 % | 0,068 ± 0,001 | 1290 | 4,8 |
| 5 % TrxR, 2,5 % DMSO | 0,035 ± 0,003 | 2521 | 2,5 |

**Fig. 6 Kinetic rates to Sec$_{ox}$. a** Homodimerization of building block **3** to (**3**)$_2$. **b** Time course for the (**3**)$_2$ formation using a concentration of 190 μM of (**3**), in 20 mM Tris buffer (pH 7.8), $T = 6$ °C, 0% (mol) Sec$_{ox}$ and 2.5% (v/v) DMSO (black dots), 5% (mol) TrxR and 2.5% (v/v) DMSO (light blue dots), 0% mol Sec$_{ox}$ and 20% (v/v) DMSO (purple dots), 1% (mol) Sec$_{ox}$ and 2.5% (v/v) DMSO (red dots), at 2.5% (mol) Sec$_{ox}$ and 2.5% (v/v) DMSO (green dots), 5% (mol) Sec$_{ox}$ and 2.5% (v/v) DMSO (blue dots) and 10% (mol) Sec$_{ox}$ and 2.5% (v/v) DMSO (orange dots). **c** Kinetic parameters. Reaction profiles were fitted to second-order reaction kinetic equations. Mean ± SD from three independent experiments. (Supplementary Fig. 38, Supplementary Methods, and Supplementary Discussion). Source data are provided as Source Data file.

**Fig. 7 Selenocystine applied to a slow DCL and a protein refolding. a** Slow thiol/disulfide exchange. Building block **5** (52,3 μM) and building blocks (**6–8**) at 104.6 μM concentration each. Tris buffer 20 mM pH 7.8 at 20% (v/v) DMSO and 6 °C. (i) 0% mol Sec$_{ox}$ as control after 192 h, (ii) 5% mol Sec$_{ox}$ (18.3 μM) after 60 h. Relative peak areas (RPAs) of the DCLs, see Supplementary Fig. 36 and Supplementary Table 2. p$K_a$ values estimated from Schrödinger Release 2020-2[54]. See Supplementary Fig. 2 for p$K_a$ calculations. **b** Kinetics for scrambled RNase A folding at pH 7.8 and room temperature. Yellow circles symbolize disulfide bonds. General conditions: scrambled RNase A (5 μM), tris buffer 100 mM, pH 7.8, 2 mM EDTA. 0.2 mM GSSG/1 mM GSH (black dots), 1 mM Sec$_{ox}$/5 mM GSH (red dots). Mean ± SD from two independent experiments. Source data are provided as Source Data file.

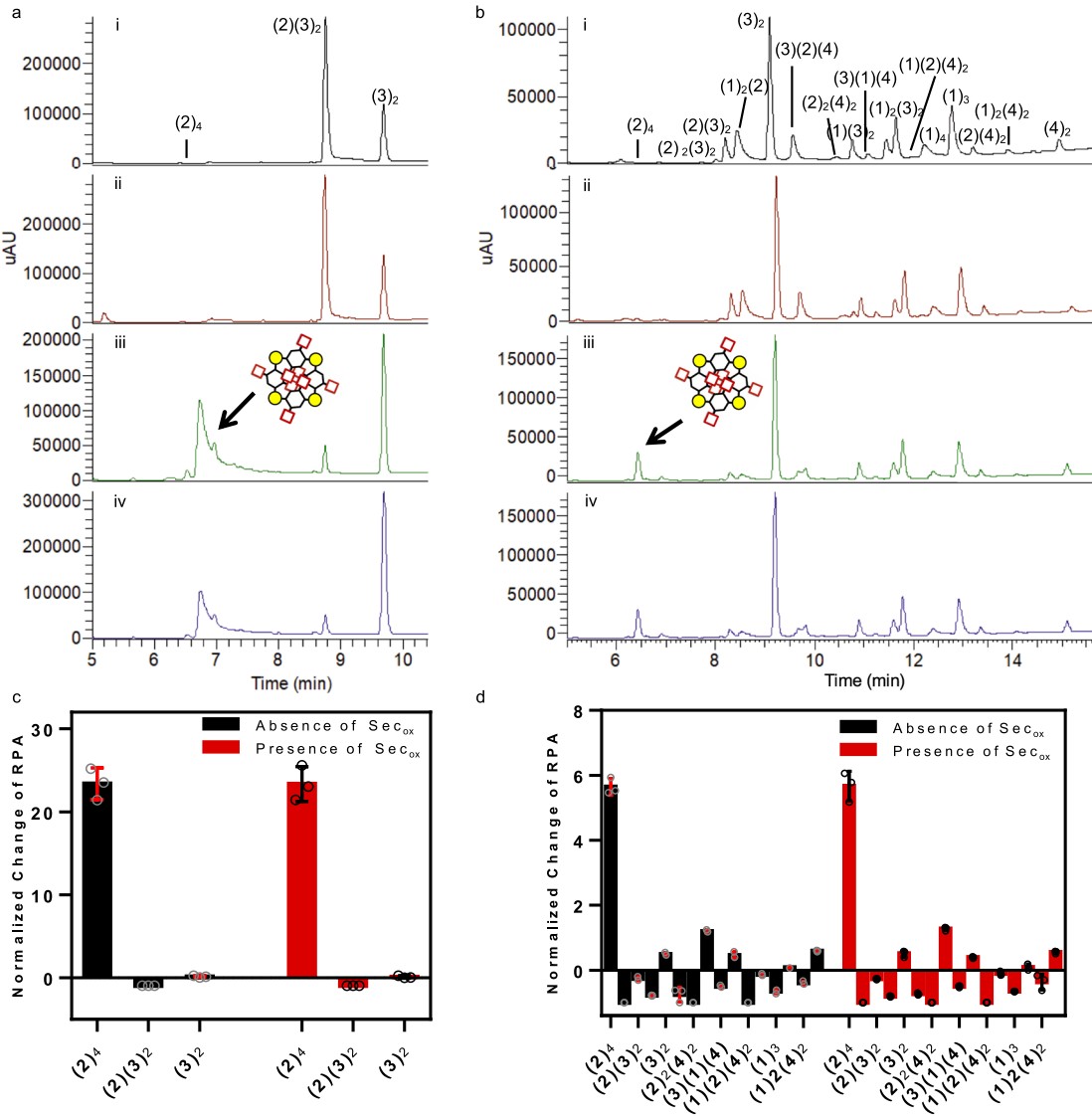

**Fig. 8 Spermine-templated DCL. a** DCL chromatograms of building block **2** (95 µM) and **3** (190 µM) in the absence and presence of spermine (28.5 µM). Yellow circle (S–S), red square (CO$_2$H), black hexagon (benzene ring). DCL conditions: Tris buffer 20 mM pH 7.8, 6 °C, 2.5% DMSO. (i) control in absence of spermine after 96 h (black), (ii) in the absence of spermine but with Sec$_{ox}$ (5% mol, 14.25 µM) 24 h (red), (iii) control in presence of spermine after 96 h (green), (iv) in the presence of spermine and 5% mol Sec$_{ox}$ 24 h (blue). **b** DCL chromatograms of building blocks **1**–**2** (95 µM each) and **3**–**4** (190 µM each), in the absence and presence of spermine (57 µM), Sec$_{ox}$ (5% mol, 28.5 µM), Tris buffer (20 mM, pH 7.8), $T = 6$ °C, 2.5% (v/v) DMSO. (i) control in absence of spermine 96 h (black), (ii) in the absence of spermine but with Sec$_{ox}$ (5%, mol 28.5 µM) 24 h (red), (iii) control in presence of spermine 96 h (green), (iv) in the presence of spermine and 5% mol Sec$_{ox}$ 24 h (blue). **c** Spermine template effect measured by the normalized change of RPA in the presence and absence of Sec$_{ox}$ for building blocks **2** and **3**. **d** Spermine template effect measured by the normalized change of RPA in the presence and absence of Sec$_{ox}$ for building blocks **1**–**4**. See Supplementary Tables 3–6. Mean ± SD from three independent experiments.

the amplification of different species and getting the same results as the non-catalyzed DCLs (Supplementary Figs. 44, 45, Supplementary Tables 7–10, and Supplementary Methods).

**Sec$_{ox}$ in a glucose oxidase-directed DCC.** Having established that Sec$_{ox}$ is an effective promoter for thiol/disulfide exchange, we next studied the introduction of the enzyme glucose oxidase (GO$x$) to the DCL. Glucose oxidase ($\beta$-D-glucose:oxygen-oxidoreductase, EC 1.1.3.4) from *Aspergillus niger* is a flavoprotein that catalyzes the oxidation of $\beta$-D-glucose to D-glucono-$\delta$-lactone and hydrogen peroxide, using oxygen as an electron acceptor. GO$x$ is well suited for DCC exploration, it is a robust and well-characterized protein with two identical subunits (molecular weight: 160 KDa)[47–49]. Besides, there are relatively few ligands reported for GO$x$ being one of the most widely used enzymes.

DCC experiments involving thiols **1**–**4** were set up in the absence and the presence of GO$x$ (Fig. 9). The stability of the protein under the experimental conditions (DMSO, Sec$_{ox}$ tolerance, and stability over time) was tested using the fluorescence technique (Supplementary Figs. 46, 47 and Supplementary Methods).

The equilibration of the DCL was completed after 24 h (Fig. 9) and the oligomers were identified by HPLC-MS (Supplementary Figs. 4–20). The precise composition of the DCL (with and without GO$x$), was assessed by measuring the relative peak area (RPA). The normalized change of RPA was used to quantify the protein influence in the final DCL showing the amplification of the dimer of compound **4**, that is (**4**)$_2$ (Fig. 9b). Control experiments were set up to verify that the amplification of the dimer was not due to unspecific binding. Firstly the GO$x$ was

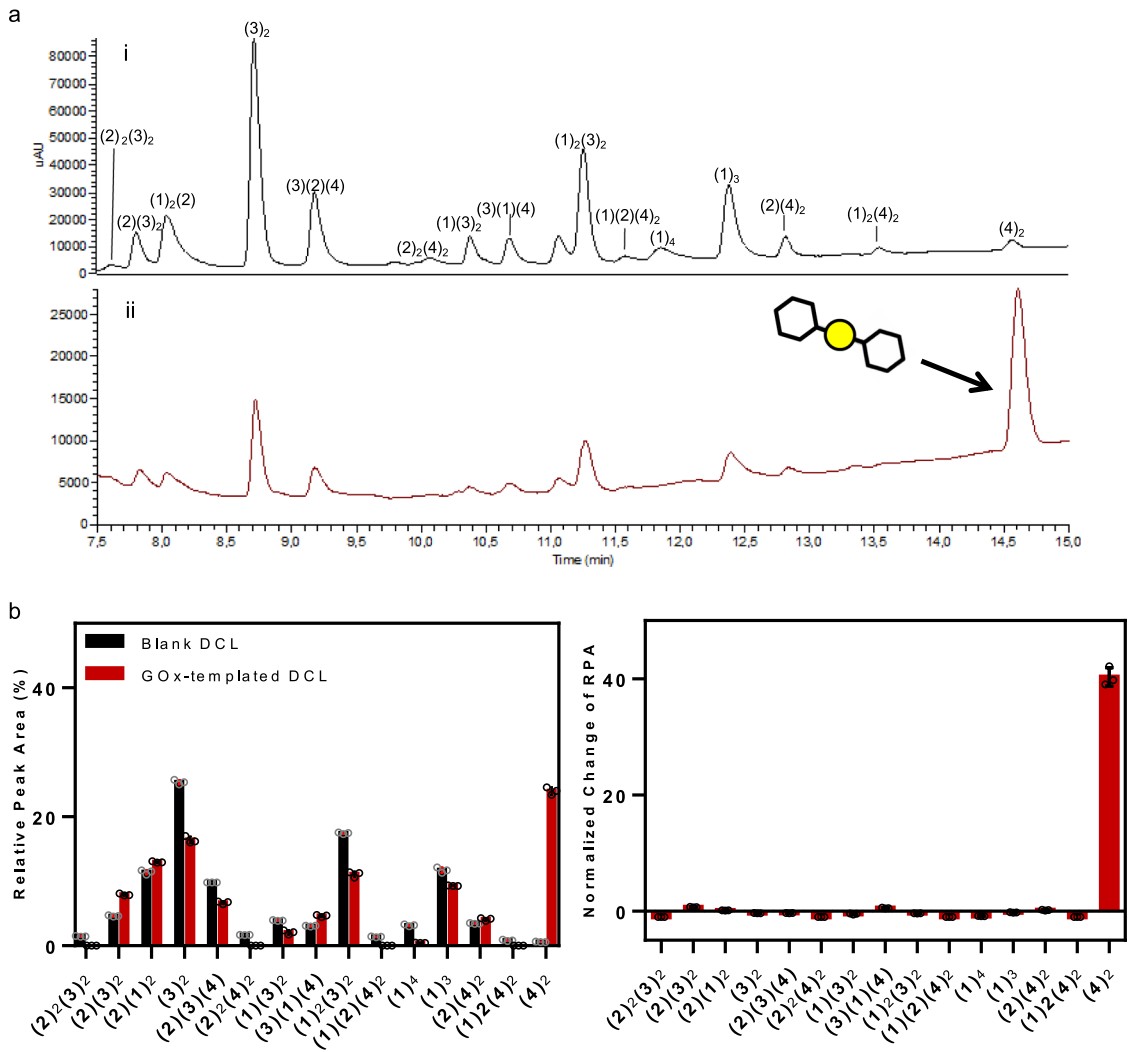

**Fig. 9 GOx-DCL by HPLC.** Conditions: building blocks **1–2** (95 µM each) and **3–4** (190 µM each), Sec$_{ox}$ (5% mol, 28.5 µM), GOx (57 µM), Tris buffer (20 mM, pH 7.8), $T = 6 \,°C$, 2.5% (v/v) DMSO. Yellow circle (S–S), black hexagon (benzene ring). **a** DCL chromatograms after 24 h (i) in absence, blank (ii) and presence of GOx, templated. **b** GOx templated effect relative peak area (RPA) and normalized change of RPA in the presence and absence of Sec$_{ox}$ (Supplementary Tables 11 and 12). DCC experiments were carried out in triplicate. Mean ± SD.

replaced by the bovine serum albumin (BSA). A stability study of BSA under the DCL conditions was performed using a fluorescence technique confirming the BSA stability in the DCL experiment (Supplementary Fig. 48)[50]. The BSA-templated DCL did not amplify (**4**)$_2$, but it modified the DCL distribution as it was previously reported for thiol–disulfide exchange (Supplementary Fig. 49 and Supplementary Methods)[51]. Therefore, we performed the DCL in the presence of a large excess of DTNB (5,5′-dithiobis-2-nitrobenzoic acid), an inhibitor of GOx, which prevented the amplification of any other ligands except DTNB (Supplementary Fig. 50), indicating that an inhibited GOx cannot influence the DCL equilibrium composition (**4**)$_2$.

**Glucose oxidase activity inhibited by compound (4)$_2$.** The amplified compound (**4**)$_2$ was isolated and its activity towards GOx tested (Fig. 10). Affinity was measured by fluorescence-based experiments (Supplementary Fig. 52 and Supplementary Methods). The natural ligand for GOx, β-D-glucose (Fig. 10a), showed an apparent dissociation constant of $1.10 \pm 0.02$ mM similar to those reported for apo and holo GOx forms ($K_d' \approx 10$ mM)[52]. DTNB was chosen as an inhibitor reference compound. Compound (**4**)$_2$ with $K_d' = 110 \pm 10$ µM,

presented affinity values similar to DTNB ($K_d' = 64 \pm 2$ µM). Glucose oxidase activity was evaluated by the horseradish-peroxidase system using o-dianisidine as a chromogen[53]. Michaelis–Menten parameters $K_M$ and $V_{max}$ and the inhibition constants $K_i$ were calculated for both inhibitors (Fig. 10c). Control experiments provided glucose conversion values of $K_M = 15 \pm 2$ mM and $V_{max} = 0.14 \pm 0.01$ min$^{-1}$ [53].

The results from Lineweaver–Burk plots determined that (**4**)$_2$ and DTNB are non-competitive inhibitors (Fig. 10b). The $K_i$ values of (**4**)$_2$ ($K_i = 1.7 \pm 0.2$ µM) and DTNB ($K_i = 0.20 \pm 0.02$ µM) were obtained using different inhibitor concentrations and the data collected were fitted to a model of non-competitive inhibitor using the program Prism 8.3.4. Furthermore, IC$_{50}$ values were performed at a final concentration of 10 mM of the substrate in presence of different concentrations of the inhibitors (1.4 nM to 2 mM). The results suggested that compound (**4**)$_2$ is a less potent inhibitor compared to DTNB.

## Discussion

Dynamic combinatorial chemistry applied to biological systems requires that the reversible chemistry of choice can be performed in practical timescales and under physiological conditions. Even

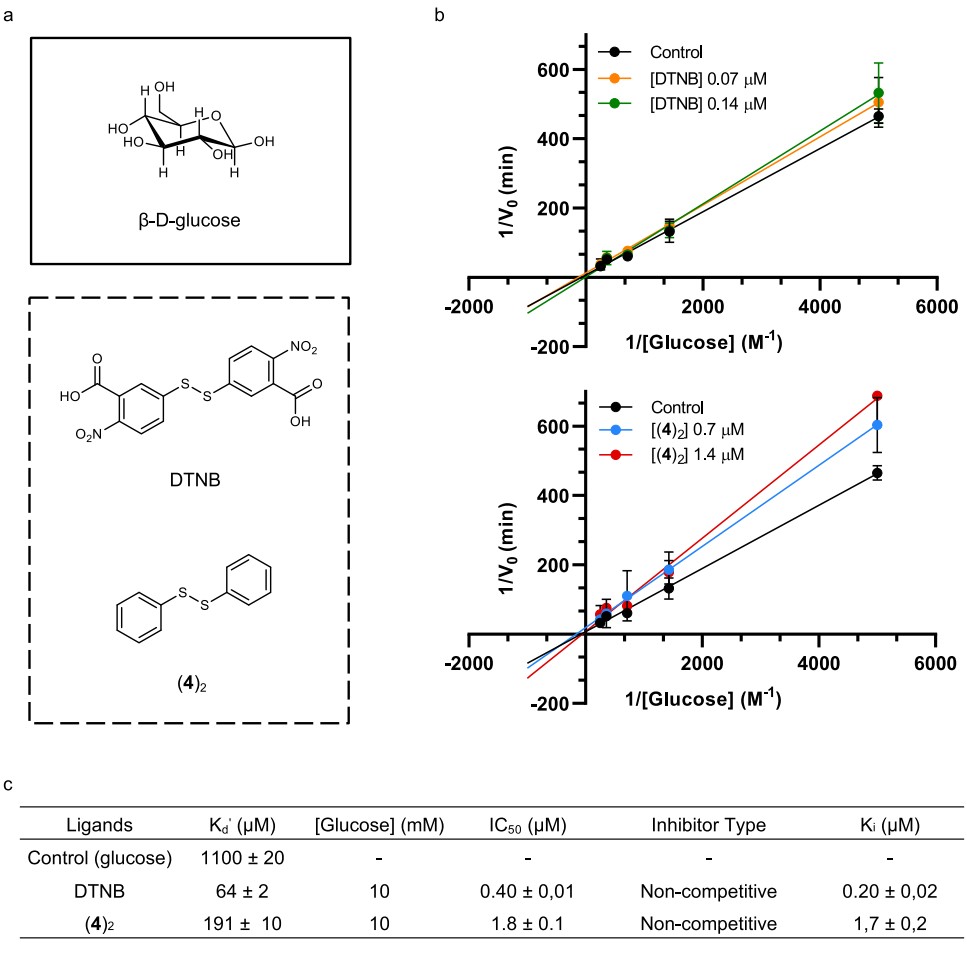

**Fig. 10 Enzyme activity and binding assays. a** Structures of substrate $\beta$-D-glucose and the studied inhibitors DTNB and $(\mathbf{4})_2$ of glucose oxidase. **b** Lineweaver–Burk plots[57] of DTNB and $(\mathbf{4})_2$. Conditions: [glucose] from 0.2 to 4 mM, 37 °C at 50 mM sodium acetate buffer pH 5.15, [DTNB] 0.07 and 0.14 µM, $[(\mathbf{4})_2]$ 0.7 and 1.4 µM. Mean ± SD from four independent experiments. **c** Summary of the binding affinity and enzymatic activity parameters (Supplementary Methods). Source data are provided as Source Data file.

| Ligands | $K_{d}'$ (µM) | [Glucose] (mM) | $IC_{50}$ (µM) | Inhibitor Type | $K_i$ (µM) |
|---|---|---|---|---|---|
| Control (glucose) | 1100 ± 20 | - | - | - | - |
| DTNB | 64 ± 2 | 10 | 0.40 ± 0,01 | Non-competitive | 0.20 ± 0,02 |
| $(\mathbf{4})_2$ | 191 ± 10 | 10 | 1.8 ± 0.1 | Non-competitive | 1,7 ± 0,2 |

though thiol–disulfide exchange is one of the most popular chemistries for DCC as it proceeds smoothly at neutral or slightly basic pH, it usually needs long equilibration times to reach its final composition.

Inspired by the way the natural enzymes thioredoxin reductases accelerate the rate of thiol–disulfide reactions, we report the use of the biocompatible selenocystine as a catalyst for a number of thiol–disulfide exchange reactions. The operating principle of this chemistry is that selenenylsulfide intermediates are more reactive towards thiolate nucleophilic additions, thus accelerating the exchange between the different species.

5% (mol) $Sec_{ox}$ is found to be optimal to reach DCL equilibration in 24 h. To study the catalytic performance of $Sec_{ox}$, kinetic studies on the homodimerization of compound **3** are performed. We show that the addition of 5% (mol) selenocystine produces an 18-fold acceleration on the homodimerization reaction of compound **3** over the uncatalyzed reaction. The use of DMSO as a co-solvent is known to accelerate the disulfide exchange. Nevertheless, the catalytic efficiency of 5% (mol) $Sec_{ox}$ clearly outcompetes that of 20% (v/v) DMSO. Moreover, $Sec_{ox}$ is catalytically more effective than selenoenzyme TrxR 2 under identical conditions. We have shown that the presence of selenocystine produced a positive effect on the thiol/disulfide exchange in a dynamic combinatorial library, being four times faster than conventional DCC even at low temperatures and physiological pH.

To broaden its scope of application as a reagent in biological redox events, the performance of $Sec_{ox}$ in the refolding of scrambled RNase A is compared to the standard glutathione redox buffer. The refolding rate increases five times by replacing GSSG with $Sec_{ox}$. Unlike GSSG, $Sec_{ox}$ is highly active even at acidic pH (5.4), expanding the range of conditions for protein refolding.

Selenocystine has proven not to interfere with DCC target-driven self-assembly processes (casting and molding) using spermine, spermidine, NADPH, and the enzyme glucose oxidase as templates. In spermine-templated DCL the equilibration time is reduced four times using 5% (mol) $Sec_{ox}$ compared to the DCL control and the cyclic tetramer $(\mathbf{2})_4$ is amplified as expected in different DCLs. The same catalytic effect is observed with the spermidine and NADPH as templates. Glucose oxidase (GO*x*) has confirmed to be an excellent DCL template directing the library to the synthesis of compound $(\mathbf{4})_2$. Accordingly, this compound presents affinity values in the micromolar range similar to the reference GO*x* inhibitor DTNB. As an additional proof for specific binding of $(\mathbf{4})_2$ to GO*x*, glucose oxidase activity is evaluated and results suggest that compound $(\mathbf{4})_2$ is a non-competitive inhibitor with a moderate inhibitory activity compared to the reported inhibitor DTNB.

In summary, the use of selenocystine as a catalyst broadens the scope of dynamic combinatorial chemistry in biological environments. It accelerates the DCLs equilibration time even at low

temperatures without interfering with the templating effect. Furthermore, selenocystine has proven its effectiveness as a catalyst promoting the formation of the right disulfide bonds in protein folding expanding the pH range conditions.

## Methods

**$Cys_{ox}$ and $Sec_{ox}$ DCL study.** Control DCL of dithiols 1–2 ($2 \times 1.8$ µL, 25 mM, $4.5 \cdot 10^{-8}$ mol per monomer, DMSO) and monothiols 3–4 ($2 \times 1.8$ µL, 50 mM, $9.0 \cdot 10^{-8}$ mol per monomer, DMSO), DMSO (4.5 µL) in 20 mM Tris buffer pH 7.8 (466 µL). DMSO percentage is 2.5% v/v. $Sec_{ox}$-catalyzed DCL: Dithiols 1–2 ($2 \times 1.8$ µL, 25 mM, $4.5 \cdot 10^{-8}$ mol per monomer, DMSO) and monothiols 3–4 ($2 \times 1.8$ µL, 50 mM, $9.0 \cdot 10^{-8}$ mol per monomer, DMSO), $Sec_{ox}$ (1.8 µL, 50 mM, $9.0 \cdot 10^{-8}$ mol per monomer, DMSO with 4% (v/v) 1 M NaOH). DMSO (2.7 µL) in 20 mM Tris buffer pH 7.8 (466 µL). $Cys_{ox}$-catalyzed DCL: dithiols 1–2 ($2 \times 1.8$ µL, 25 mM, $4.5 \cdot 10^{-8}$ mol per monomer, DMSO) and monothiols 3–4 ($2 \times 1.8$ µL, 50 mM, $9.0 \cdot 10^{-8}$ mol per monomer, DMSO), $Cys_{ox}$ (1.8 µL, 50 mM, $9.0 \cdot 10^{-8}$ mol per monomer, $H_2O$), DMSO (4.5 µL) in 20 mM Tris buffer pH 7.8 (464.2 µL). DCLs were analyzed by HPLC-MS until equilibration without stirring at 6 °C.

**Percentage (% mol) of $Sec_{ox}$.** Control DCL. Dithiols 1–2 ($2 \times 1.8$ µL, 25 mM, $4.5 \cdot 10^{-8}$ mol per monomer, DMSO) and monothiols 3–4 ($2 \times 1.8$ µL, 50 mM, $9.0 \cdot 10^{-8}$ mol per monomer, DMSO), DMSO (4.5 µL) in 20 mM Tris buffer pH 7.8 (466 µL). DMSO percentage is 2.5% v/v. Regarding $Sec_{ox}$ catalyzed DCL, different stocks of $Sec_{ox}$ were carried out in order to study the percentages of $Sec_{ox}$. The general catalyzed DCL was the addition of dithiols 1–2 ($2 \times 1.8$ µL, 25 mM, $4.5 \cdot 10^{-8}$ mol per monomer, DMSO), monothiols 3–4 ($2 \times 1.8$ µL, 50 mM, $9.0 \cdot 10^{-8}$ mol per monomer, DMSO), $Sec_{ox}$ (1.8 µL, corresponding concentration, DMSO with 4% (v/v) 1 M NaOH), and DMSO (2.7 µL) in 20 mM Tris buffer pH 7.8 (466 µL). Concentration according to the percentage of $Sec_{ox}$: 10% mol—15 mM stock ($2.7 \cdot 10^{-8}$ mol), 5% mol—7.5 mM stock ($1.35 \cdot 10^{-8}$ mol), 2.5% mol—3.75 mM ($6.7 \cdot 10^{-9}$ mol), 1% mol—1.5 mM stock ($2.7 \cdot 10^{-9}$ mol). The DCLs were prepared at 6 °C without stirring and analyzed in several hours until the complete stabilization.

**Kinetic studies.** The reaction started by the addition of the thiol 3 (3.8 µL, 50 mM, $1.9 \cdot 10^{-7}$ mol, in DMSO) over the mixture of $Sec_{ox}$ (3.8 µL, at different concentrations according to the percentage, in DMSO with 4% (v/v) 1 M NaOH) or DMSO in control DCL (3.8 µL), DMSO (17.4 µL) and buffer Tris 20 mM pH 7.8 (975 µL) up to a total volume of 1 mL, and a final percentage of 2.5% (v/v) DMSO. $T = 6$ °C without stirring. Concentration according to the percentage of $Sec_{ox}$: 10% mol—5 mM stock ($1.9 \cdot 10^{-8}$ mol), 5% mol—2.5 mM stock ($9.5 \cdot 10^{-9}$ mol), 2.5% mol—1.25 mM ($4.75 \cdot 10^{-9}$ mol), 1% mol—0.5 mM stock ($1.9 \cdot 10^{-9}$ mol). For the 20% (v/v) DMSO-reaction, 3 was added over the solution of DMSO (196.2 µL) and buffer Tris 20 mM pH 7.8 (800 µL) following the same parameters described above. Besides, for TrxR-catalyzed reaction, 3 (3.8 µL, 50 mM, $1.9 \cdot 10^{-7}$ mol, in DMSO) was added over a solution of TrxR (40 µL, 237.5 µM, $9.5 \cdot 10^{-9}$ mol, in buffer Tris), DMSO (21.2 µL) and buffer Tris 20 mM pH 7.8 (935 µL). For the data treatment, see Supplementary Methods and Supplementary Discussion.

**Slow DCL system.** Control DCL. Dithiol 5 (1.0 µL, 25 mM, $2.5 \cdot 10^{-8}$ mol, DMSO) and monothiols 6–8 ($3 \times 1.0$ µL, 50 mM, $5 \cdot 10^{-8}$ mol per monomer, DMSO), DMSO (91.6 µL) in 20 mM Tris buffer pH 7.8 (382 µL), affording a final DMSO percentage of 20% (v/v). $Sec_{ox}$-catalyzed: Dithiol 5 (1.0 µL, 25 mM, $2.5 \cdot 10^{-8}$ mol, DMSO) and monothiols 6–8 ($3 \times 1.0$ µL, 50 mM, $5 \cdot 10^{-8}$ mol per monomer, DMSO), $Sec_{ox}$ (1.0 µL, 8.75 mM, $8.75 \cdot 10^{-9}$ mol, DMSO with 4% (v/v) 1 M NaOH), DMSO (90.6 µL) in 20 mM Tris buffer pH 7.8 (382 µL), DMSO is 20% (v/v) due to the solubility issues, at 6 °C and without stirring. The DCLs were analyzed until stabilization.

**RNase A refolding by $Sec_{ox}$.** The activity of folded scrambled RNase was determined by the selective hydrolysis of cCMP. The folding experiment was carried out by the addition of scrambled RNase A at a final concentration of 5 µM in presence of two redox pairs, GSSG/GSH (0.2 mM/1 mM) and $Sec_{ox}$/GSH (1 mM/5 mM) in buffer 20 mM Tris 2 mM EDTA pH 7.8 at room temperature. The concentration of the properly folded protein was determined by the calibration curve based on the initial velocity vs. the concentration. Aliquots (30 µL) of the refolding solution were withdrawn to measure the hydrolysis of cCMP for 2 min at 25 °C spectrophotometrically (292 nm). The experiment was evaluated for 3 days at prescribed times. Refolding experiments at pH 5.4 are detailed in Supplementary Methods and Supplementary Discussion.

**$Sec_{ox}$ catalyzing Spermine-directed DCLs**

*Methodology of two building blocks DCLs.* Control DCL: dithiol 2 (1.8 µL, 25 mM, $4.5 \cdot 10^{-8}$ mol, DMSO) and monothiol 3 (1.8 µL, 50 mM, $9.0 \cdot 10^{-8}$ mol, DMSO), DMSO (8.1 µL) in 20 mM tris buffer pH 7.8 with 2.5% (v/v) DMSO. The DCL was stabilized at 96 h. Control DCL-$Sec_{ox}$ catalyzed: dithiol 2 (1.8 µL, 25 mM, $4.5 \cdot 10^{-8}$ mol, DMSO) and monothiol 3 (1.8 µL, 50 mM, $9.0 \cdot 10^{-8}$ mol, DMSO), $Sec_{ox}$ (1.8 µL, 3.75 mM, $6.75 \cdot 10^{-9}$ mol, DMSO with 4% (v/v) 1 M NaOH), DMSO (6.3 µL) in

20 mM tris buffer pH 7.8 (466 µL) with 2.5% (v/v) DMSO. The mixture was stabilized after 24 h. Spermine-directed DCL: dithiol 2 (1.8 µL, 25 mM, $4.5 \cdot 10^{-8}$ mol, DMSO) and monothiol 3 (1.8 µL, 50 mM, $9.0 \cdot 10^{-8}$ mol, DMSO), DMSO (8.1 µL), Spermine solution in 20 mM tris buffer pH 7.8 (5 µL, 2.7 mM, $1.35 \cdot 10^{-8}$ mol), and 20 mM tris buffer pH 7.8 with 2.5% (v/v) DMSO (461 µL). The DCL was stabilized at 96 h. Spermine-directed-$Sec_{ox}$ catalyzed DCL: Dithiol 2 (1.8 µL, 25 mM, $4.5 \cdot 10^{-8}$ mol, DMSO) and monothiol 3 (1.8 µL, 50 mM, $9.0 \cdot 10^{-8}$ mol, DMSO), $Sec_{ox}$ (1.8 µL, 3.75 mM, $6.75 \cdot 10^{-9}$ mol, DMSO with 4% (v/v) 1 M NaOH), DMSO (6.3 µL), Spermine solution in 20 mM tris buffer pH 7.8 (5 µL, 2.7 mM, $1.35 \cdot 10^{-8}$ mol), and 20 mM tris buffer pH 7.8 with 2.5% (v/v) DMSO (461 µL). The DCL was analyzed in 96 h after equilibration. DCLs were performed without stirring at 6 °C and analyzed by HPLC-MS.

*Methodology of four building blocks DCLs.* Control DCL: dithiols 1–2 ($2 \times 1.8$ µL, 25 mM, $4.5 \cdot 10^{-8}$ mol per monomer, DMSO), monothiols 3–4 ($2 \times 1.8$ µL, 50 mM, $9.0 \cdot 10^{-8}$ mol per monomer, DMSO), DMSO (4.5 µL) in 20 mM tris buffer pH 7.8 with 2.5% (v/v) DMSO. The mixture was stabilized in 96 h. Control DCL-$Sec_{ox}$ catalyzed: dithiols 1–2 ($2 \times 1.8$ µL, 25 mM, $4.5 \cdot 10^{-8}$ mol per monomer, DMSO) and monothiols 3–4 ($2 \times 1.8$ µL, 50 mM, $9.0 \cdot 10^{-8}$ mol per monomer, DMSO), $Sec_{ox}$ (1.8 µL, 7.5 mM, $1.35 \cdot 10^{-8}$ mol, DMSO with 4% (v/v) 1 M NaOH), DMSO (2.7 µL) in 20 mM tris buffer pH 7.8 (466 µL) with 2.5% (v/v) DMSO. The mixture was stabilized after 24 h. Spermine-directed DCL: dithiols 1–2 ($2 \times 1.8$ µL, 25 mM, $4.5 \cdot 10^{-8}$ mol per monomer, DMSO) and monothiols 3–4 ($2 \times 1.8$ µL, 50 mM, $9.0 \cdot 10^{-8}$ mol per monomer, DMSO), DMSO (4.5 µL), Spermine solution in 20 mM tris buffer pH 7.8 (10 µL, 2.7 mM, $2.7 \cdot 10^{-8}$ mol) and 20 mM tris buffer pH 7.8 with 2.5% v/v DMSO (456 µL). It was stabilized in 24 h. Spermine-directed-$Sec_{ox}$catalyzed DCL: dithiols 1–2 ($2 \times 1.8$ µL, 25 mM, $4.5 \cdot 10^{-8}$ mol per monomer, DMSO), monothiols 3–4 ($2 \times 1.8$ µL, 50 mM, $9.0 \cdot 10^{-8}$ mol per monomer, DMSO), $Sec_{ox}$ (1.8 µL, 7.5 mM, $1.35 \cdot 10^{-9}$ mol, DMSO with 4% (v/v) 1 M NaOH), DMSO (2.7 µL), Spermine solution in 20 mM tris buffer pH 7.8 (10 µL, 2.7 mM, $2.7 \cdot 10^{-8}$ mol) and 20 mM tris buffer pH 7.8 with 2.5% v/v DMSO (456 µL). DCL was equilibrated in 96 h. DCLs were performed without stirring at 6 °C and analyzed by HPLC-MS.

See Supplementary Methods for more templated-DCLs (e.g., Spermidine and NADPH).

**Glucose oxidase-directed DCL.** Blank DCL: Dithiols 1–2 ($2 \times 1.8$ µL, 25 mM, $4.5 \cdot 10^{-8}$ mol per monomer, DMSO) and monothiols 3–4 ($2 \times 1.8$ µL, 50 mM, $9.0 \cdot 10^{-8}$ mol per monomer, DMSO), $Sec_{ox}$ (1.8 µL, 7.5 mM, $1.35 \cdot 10^{-8}$ mol, DMSO), DMSO (2.7 µL) in 20 mM tris buffer pH 7.8 with 2.5% (v/v) DMSO (466 µL). GO*x*-directed DCL: Dithiols 1–2 ($2 \times 1.8$ µL, 25 mM, $4.5 \cdot 10^{-8}$ mol per monomer, DMSO) and monothiols 3–4 ($2 \times 1.8$ µL, 50 mM, $9.0 \cdot 10^{-8}$ mol per monomer, DMSO), $Sec_{ox}$ (1.8 µL, 7.5 mM, $1.35 \cdot 10^{-8}$ mol, DMSO with 4% (v/v) 1 M NaOH), DMSO (2.7 µL), GO*x* (135 µM, 200 µL, $2.7 \cdot 10^{-8}$ mol, 10% mol) in 20 mM tris buffer pH 7.8 with 2.5% (v/v) DMSO (266 µL). The DCL was stabilized for 24 h at 6 °C. Then, GO*x* was removed by ultracentrifugation through an Amicon ultra-filter (100 KDa). HPLC analysis was performed.

Supplementary Methods: BSA-directed DCL and GO*x*-directed DCL in presence of DTNB.

**Reversibility study.** See Supplementary Methods and Supplementary Discussion.

**Synthesis of building blocks 1, 2, and 5.** See Supplementary Methods and Supplementary Discussion.

**Synthesis of compounds (3)$_2$ and (4)$_2$.** See Supplementary Methods and Supplementary Discussion.

**Protein stability experiments.** See Supplementary Methods.

**Fluorescence emission experiments.** See Supplementary Methods.

**GO*x* activity assays.** Enzymatic assay to determine Michaelis–Menten constant $K_m$ and maximum velocity $V_{max}$, $IC_{50}$, and inhibition constant $K_i$ of the ligands in presence of compounds were performed by glucose oxidase-horseradish peroxidase-coupled system using *o*-dianisidine as chromogen, with a 2.5% (v/v) of DMSO.

Michaelis–Menten parameters were set up using different final concentrations of glucose from 0.2 to 4 mM. Glucose stocks from 0.5 to 10 mM (100 µL, 50 mM sodium acetate buffer pH 5.1), *o*-dianisidine (127.8 µL, 3.5 mM, MilliQ water), DMSO (6.2 µL), horseradish peroxidase (8 µL, 207 µM, 50 mM sodium acetate buffer pH 5.1) and GO*x* (8 µL, 42.7 nM, 50 mM sodium acetate buffer pH 5.1) were added to a final volume of 250 µL in a 2.5% (v/v) DMSO. GO*x* activity was measured spectrophotometrically at 500 nm 20 min at 37 °C. The data were fitted to a Lineweaver–Burk model.

$IC_{50}$ was determined using different concentrations of inhibitors DTNB and (4)$_2$. The solution containing glucose (100 µL, 25 mM, 50 mM sodium acetate

buffer pH 5.1), o-dianisidine (127.8 μL, 3.5 mM, MilliQ H$_2$O), DTNB, or (**4**)$_2$ at different concentrations (6.2 μL, from 1.4 nM to 2 mM, DMSO), horseradish peroxidase (8 μL, 207 μM, 50 mM sodium acetate buffer pH 5.1) and GO*x* (8 μL, 42.7 nM, 50 mM sodium acetate buffer pH 5.1) was analyzed spectrophotometrically at 500 nm in a plate reader, 20 min at 37 °C. The data were fitted to a dose–response model for three parameters using Prism 8.3.4.

Inhibition constant $K_i$ was determined by the use of several ligands and substrate concentrations. Glucose (100 μL, 0.5−10 mM, in 50 mM sodium acetate buffer pH 5.1), o-dianisidine (127.8 μL, 3.5 mM, MilliQ H$_2$O), DMSO, several concentrations of DTNB (35−140 nM) or (**4**)$_2$ (0.07−1.4 μM) in DMSO (6.2 μL), horseradish peroxidase (8 μL, 207 μM, 50 mM sodium acetate buffer pH 5.1) and GO*x* (8 μL, 42.7 nM, 50 mM sodium acetate buffer pH 5.1). Plates were measured spectrophotometrically at 500 nm 20 min at 37 °C. The data were fitted to a non-competitive inhibition model by Prism 8.3.4. For more details, see Supplementary Methods.

## Data availability

A reporting summary for this Article is available as a Supplementary Information file. The rest of the data are available in the Supplementary Information File. Source data are provided with this paper.

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

## Acknowledgements

We thank Prof. J. Elguero (IQM-CSIC) and Dr. G. Jiménez-Osés (CICbioGUNE) for helpful discussions on the paper and Dr. F. J. Medrano (CIB-CSIC) for his assistance with the folding studies. This work was supported by the Spanish Ministry of Science and Innovation with Grant PID2019-108587RB-I00 (R.P.F.)

## Author contributions

A.C.-M. and R.P.-F. designed the experiments. A.C.-M. performed the DCC, enzymatic assays, fluorescence, and folding studies. R.P.-F. wrote the paper.

## Competing interests

The authors declare no competing interests.
