## [Peer Review File · Nature Communications]

Reviewer #1 (Remarks to the Author):

This is the review-report for:

Biomimetic selenocystine based dynamic combinatorial chemistry for thiol-disulfide exchange
by Andrea Canal-Martín and Ruth Pérez-Fernández

The article deals with a new approach to dynamic combinatorial chemistry (DCC) using reversible disulfides by using bio-mimetic catalysis with selenocystine derivatives under physiological conditions. By applying the nature-inspired method, the authors wanted to solve the known problem of long equilibration times of disulfide dynamic combinatorial libraries (DCL). The authors present a model DCL containing numerous macrocyclic and linear products based on four simple and known mono and difunctional thiophenols functionalized with carboxylic groups. The next step is the optimization of selenocystine catalysis in various physiological pH buffers and different temperature ranges. Then, the authors try to reach some control on this DCL by the templation experiments with spermine and enzymatic templation with Glucose Oxidase, which in both cases result in significant changes in the DCL composition.

This is a meticulously done work. Experimental part is almost entirely based on LC-MS analysis. Qualitative analysis of such a complex DCL containing almost 20 dynamic products must have been a tedious work. The authors also used fluorescence spectroscopy to determine the dissociation constant of the dynamic disulfide product to the Glucose Oxidase. The experimental part has been described accurately which will certainly facilitate future reproduction of the presented results.

The application of selenocystine and its derivatives for disulfide catalysis has been already reported, also in terms of dynamic disulfide chemistry (which the authors fairly admitted). The novelty here is, therefore, the use of this effect in a more complex DLC and various conditions (pH, temp.). However, the authors do not explain why they choose exactly those building blocks 1-4, or why they decided to work on such a complicated library. The templation effect of polyamines (in presence of carboxylic groups) is widely known in this field, so the fact that the addition of spermine changed the composition of the carboxylic-functionalized DCL is not a surprise at all. Why more different templates were not used and compared? Even if the spermine experiment was to exclude a possible conflict between the templation effect and selenocystine catalysis, confirming this with only one template is insufficient. The authors also do not explain why they chose spermine for a template and do not discuss the reasons for topological changes in dynamic products caused by spermine (linear to macrocyclic). The enzymatic templation in terms of disulfide DCC is indeed very interesting, but I don't understand why the authors chose the glucose oxidase for this purpose. If the inspiration for the work was thioredoxin reductase, then it would be really interesting to compare the catalytic activity of this enzyme with the bio-mimic selenocystine system in equilibrating this DCL.

Determining the dissociation constant of the enriched dynamic product (4)₂ relative to glucose (the native substrate for glucose oxidase) is certainly interesting, but does not seem to have a logical connotation to the rest of the work. It requires further clarification.

From my point of view, the only interesting novelty in this paper is the development of a quite useful methodology for the catalysis of dynamic disulfide exchange with selenocystine derivatives under physiological conditions. It will certainly be significant facilitation for all researchers who struggle in this field of chemistry. In my opinion, the paper should be revised and submitted again to a more specialised journal.

Comments:

I am not the enthusiast of the estimated pKa data from Marvin Sketch given in Figure 3b-c. There is no reference about the methods used by this software. If the authors believe that comparing the

pKa values are crucial for the paper, they should provide experimental data for these compounds. In the experimental section about experiments with glucose oxidase (page 9), DMSO appears unexpectedly in the reaction mixture (for protein stabilisation). The authors do not consider the effect of DMSO on thiol oxidation, although it is widely known and cited in the introduction (ref. 13). Paper is overloaded by unnecessary curiosities, such as: mentioning glucose oxidase applications (refs. 33-35 and refs. 44-46) and mentioning the biological role of spermine (which is doubled by the way, ref. 31 and refs. 38-40).

The yellow circles symbolising the -SH groups in Figures 3 and 6 are not very visible. The contrast of yellow blocks should be increased.

Figure 2 is unclear. The bonds and electric charges symbols are almost invisible/overlay. This should be corrected.

There is no information in Supplementary Figures 2-20 about, if is the isolated product chromatogram or extracted ion chromatogram and about formula of observed mass ions. The structural formula, exact mass, and calculated/found ion adduct is obligatory here.

In Supplementary Information, the authors claimed that "All chemicals and solvents were used from commercial sources such as [...]" (page 12), next they said, that components 1 and 2 were synthesised according to previously reported method (page 14). There is no characterisation of them (NMR or MS).

In Supplementary Information, there is a compound (6)₂ containing the unlabeled component 6 (page 14-15). I think it should be (3)₂ according to its name 3,3'-disulfanediyldibenzoic acid (page 15).

Reviewer #2 (Remarks to the Author):

This work reports on the clever use of selenocystine, the oxidized form of selenocysteine, for the promotion of fast disulfide exchange at low temperature, with potential applications in dynamic combinatorial chemistry. The authors optimized the concentration of the diselenide catalyst and showed the potential application of the method to two already known dynamic combinatorial libraries: the optimization of a receptor for spermine and the identification of a potential ligand for glucose oxidase enzyme. The work is mainly methodological and would have great potential for the researchers working in the field of DCC for biological environments. Despite the study seems to be carefully planned and executed, there are several technical issues that must be answered and addressed for the unambiguous extraction of the corresponding conclusions. Moreover, I have some doubts about the suitability of the presentation of the work for a multidisciplinary scientific audience. I am also suggesting some tips to make the work more appealing for the readership of Nature Communications. Overall, I believe this contribution could become suitable for Nature Communications after the main concerns have been addressed.

General suggestions:

1) As I said before, the results will be highly appealing for a reduced scientific community, working in the DCC field. It shows an apparently practical methodology for its use in specific applications. However, the work would be even more attractive for the general scientific community if the authors can show the generalized application of the method to the formation and exchange of

disulfide bonds in, for instance, the correct disulfide bond in synthetic peptides containing Cys, or in the folding of a small protein.

2) Alternative to the previous point, if the methodology can be used in systems where the slow disulfide oxidation and exchange are impractical but becomes fast and efficient with the selenocystine, the relevance of the proposed method would be higher. Despite I understand that these two points (1 and 2) are not essential for acceptance in Nature Communications, I believe they would make the paper more attractive, thus improving the potential impact of the work.

3) The authors argued that they were inspired by a natural enzyme, but the similarity is rather low. The enzyme does not use diselenide bonds. They should downgrade that bio-inspiration or clearly underline the differences between the enzyme and their proposal.

4) Another fact that slightly reduces the innovative aspects of the work is the use of known dynamic combinatorial mixtures against known templates. Have the authors tried to apply it to a new non-reported system? Do the authors envision their general application?

5) Regarding the proposed mechanism through the formation of Se-S intermediate bond, do the authors have any experimental evidence of such species? A simple ESI-MS identification of this intermediate would be good enough, considering that intermediate is expected to exist in very low concentration.

Technical issues:

1) The authors said that the presence of different amounts of selenocystine does not change the DCL composition. By a close inspection of figure 5, this is unclear to me, since the relative heights of the (3)(2)(4) or the (2)4 peaks are slightly different in some experiments. Sometimes this is difficult to assess by naked eye, and the superposition of the spectra or the comparison of the integrated relative areas are more reliable (see, for instance, Figure 6).

2) In the GOx section, the amplified molecule could be a ligand for any type of hydrophobic pocket. Have the authors tested if this is the case using a dummy protein such as BSA or HSA. These could be rather conflictive since these proteins have several disulfide bonds. However, if the method also works in these cases, its impact would be higher.

3) Related to that, the fluorescence titration of the enzyme with the ligand and glucose is not very informative. First of all, the authors should mention (main article or supporting information) which method or equation they used for the fitting. Moreover, the observed binding could be unspecific in this case. Have the authors checked the GOx activity with/without the amplified ligand?

4) During the evaluation of the manuscript, I was quite surprised about the use of selenocystine in aqueous media, because this molecule is extremely insoluble in water. After the reading of the supporting information, I realized that the selenocystine was included in the reaction adding DMSO as a cosolvent. On the other hand, it is known that small amount of DMSO can also increase the disulfide exchange (as the authors cited in the introduction). Can the authors be sure that the increase in the rate of reaching the equilibrium is exclusively due to the presence of the diselenide? Do the corresponding reactions without added thiol, with Cys or with cystine also contain the same amount of DMSO as with selenocystine? In case of needed, the corresponding control experiments might be performed to confirm that the observed effects are due to the diselenide catalyst.

Other minor issues:

1) All the amino acids shown in Figure 3 have D-stereochemistry. I guess this is an error, please amend that. The same issue states for the structures in Figure 1.

2) For every mention to % of a reagent, especially in the methods part, it should be clearly stated if this is volume, weight of molar percent, and with respect to what.

3) Some of the disulfides formed in the libraries (i.e. (4)2) are expected to be poorly soluble in

buffered water. The authors must clarify if they checked that possibility and how. When quantifying species in a mixture, solubility problems must be carefully considered.

Reviewer #3 (Remarks to the Author):

In this manuscript the authors describe that the selenocysteine (oxidised diselenide form) is capable of catalysing the thiol-disulfide exchange in a dynamic combinatorial library. It is shown how a library of up to four thiols or dithiols reach equilibrium faster in the presence of selenocysteine. The same observation was made for the target-driven self-assembly in the presence of spermine or glucose oxidase (GOx).

It has previously been observed (ref 15) that selenocystamine can catalyse disulfide exchange. Most of the building blocks have been used in previous DCC systems, and spermine has also previously been used as target (Ref 41 and others). Whereas the protein templated DCC has been shown before it has not been shown with GOx.

The novelty in the MS is that selenocysteine can catalyse the disulfide exchange in a dynamic combinatorial library. This is an interesting and important discovery since one of the drawbacks of thiol/disulphide DCC is the slow rates.

The main problem of the MS is that the authors only describe the rate enhancement by comparing if the reactions have reached equilibrium after 24 h or 96 h, as characterised by HPLC. It is shown for different ratios of the catalyst, but it doesn't provide any quantitative measure of the rate acceleration. And since this is the main point of the study, I find the manuscript too premature for publication in Nature Communication. The authors should provide quantitative characterisation of the rate constants, starting with a simple homodimerisation of thiols, as it is done in ref. 15.

In addition, the authors should also refer more precisely to prior work. When describing the building blocks, they should refer to prior work by Sanders and Otto that have used the same structure. There are also more studies on the spermine templated disulphide DCC than ref. 41 (See Journal of Systems Chemistry 2013, 4, 2 and Chem. Commun., 2009, 3708–3710 for both building blocks and spermine). On the other hand, the authors can leave out some of the many references to glucose which seem unnecessary.

For the glucose templated DCC where 10% mol GOx is used, I am wondering how it can result in what seems to be approx. 20% mol of the binder (4)2, especially since it has a binding constant of 1.9 mM to GOx while the building blocks and glucose are present in micromolar concentrations. Please explain.

In the experimental section in the MS, please provide the molar amount of the chemicals in addition to the concentration and volume.

We would like to thank you for reviewing our manuscript “Biomimetic selenocystine based dynamic combinatorial chemistry for thiol-disulfide exchange”. Your inputs have been very helpful for improving the manuscript, which has been revised and completed. The modifications have been highlighted in color blue in the manuscript and added in the Supporting Information.

Point-by-point response to the concerns raised by the reviewers:

Reviewer #1.

--The application of selenocystine and its derivatives for disulfide catalysis has been already reported, also in terms of dynamic disulfide chemistry (which the authors fairly admitted). The novelty here is, therefore, the use of this effect in a more complex DLC and various conditions (pH, temp.). However, the authors do not explain why they choose exactly those building blocks 1-4, or why they decided to work on such a complicated library.

The DCL was made of dithiols and monothiols to form different architectures such as cyclic and linear oligomers, expanding the possibilities for the molecular recognition of different templates. Besides, the addition of carboxylate groups contributed to their solubility under the DCL conditions and the interaction with positively charged amines from the templates chosen through hydrogen bonding and ionic interactions. The aromatic rings may participate in hydrophobic interactions with the corresponding templates. We decided to work on a complicated library to challenge selenocystine performance in the exchange process of a rich mixture of oligomers. This information regarding the building blocks choice has been included in the manuscript.

--The templation effect of polyamines (in presence of carboxylic groups) is widely known in this field, so the fact that the addition of spermine changed the composition of the carboxylic-functionalized DCL is not a surprise at all. Why more different templates were not used and compared? Even if the spermine experiment was to exclude a possible conflict between the templation effect and selenocystine catalysis, confirming this with only one template is insufficient. The authors also do not explain why they chose spermine for a template and do not discuss the reasons for topological changes in dynamic products caused by spermine (linear to macrocyclic).

We chose the two building blocks DCL templated by spermine reported by Vial et al. as a well-studied system to test that Sec_{ox} did not interfere with the recognition process and that this DCL was efficiently equilibrated in the presence of selenocystine. The polyamine spermine was the template of the DCL made of building blocks equipped with carboxylic groups to interact through hydrogen bonding and ionic interactions with positively charged amines. Regarding the spermine example, the studies explaining how the spermine threaded through the macrocycle (2)₄, fixing the configuration of one of the four

diastereomers of (2)₄ was already described by Vial et al. We have included in the manuscript a sentence to mentioned it.

Following the referee's suggestion and to confirm the absence of conflict between selenocystine and templating effect, we have included two more templates (spermidine and NADPH) in addition to spermine (Figure 8) in our 4 building blocks DCL. The chromatograms and normalized change of relative peak areas of the DCL templated by spermidine and NADPH have been included in the Supplementary Information. The compounds amplified in our 4 thiols DCL with the different templates are: Spermine compound (2)₄, Spermidine compound (2)₄, NADPH compounds (2)(3)₂ and (1)(3)₂. Moreover, the equilibration time was reduced from four to one day in the presence of selenocystine.

--The enzymatic templation in terms of disulfide DCC is indeed very interesting, but I don't understand why the authors chose the glucose oxidase for this purpose.

A part from a variety of salts, there are few small molecules reported as inhibitors of GOx. We found challenging to find small molecules as modulators of an enzyme with such a variety of applications and market.

--If the inspiration for the work was thioredoxin reductase, then it would be really interesting to compare the catalytic activity of this enzyme with the bio-mimic selenocystine system in equilibrating this DCL.

On the revised version of the manuscript we have included the quantitative measure of the rate acceleration on the homodimerization reaction of compound 3 to (3)₂ during the first 13 hours in the presence of different percentages of selenocystine, 20% (v/v) DMSO and 5 % (mol) recombinant human TrxR 2 as the referee suggested (Figure 6). We observed that Sec_{ox} was catalytically more effective than selenoenzyme TrxR 2 under identical conditions. Indeed, the kinetic rate observed with 5% (mol) selenoenzyme resulted lower than with 1% (mol) Sec_{ox}. All the data has been incorporated to Figure 6c and Supplementary Information.

--Determining the dissociation constant of the enriched dynamic product (4)₂ relative to glucose (the native substrate for glucose oxidase) is certainly interesting, but does not seem to have a logical connotation to the rest of the work. It requires further clarification.

We have included further studies to complete the previous version of the manuscript where we only performed binding affinity assays to determine the affinity for the compounds to the protein. The fluorescence assay have guided the concentration of the inhibitors for the enzyme activity assays now included (Figure 10). In this revised version of the manuscript the glucose oxidase activity was evaluated by the horseradish-peroxidase system using o-dianisidine as a chromogen. DTNB was chosen as an inhibitor reference compound. The results suggest that compound (4)₂ and DTNB are non-

competitive inhibitors with K_i values of $(4)_2$ ($K_i = 1.7 \pm 0.2 \mu\text{M}$) and DTNB ($K_i = 0.20 \pm 0.02 \mu\text{M}$).

With these studies we have proven the affinity of the compound selected by the Glucose oxidase in the DCL (affinity binding studies) and its mode of inhibition (Glucose oxidase activity assays).

Comments:

1. I am not the enthusiast of the estimated pKa data from Marvin Sketch given in Figure 3b-c. There is no reference about the methods used by this software. If the authors believe that comparing the pKa values are crucial for the paper, they should provide experimental data for these compounds.

The Marvin sketch estimation has been removed from the manuscript. In the absence of experimental values for the referred pK_as, such values have been calculated using Schrödinger Release 2020-2. Epik, Schrödinger LLC, New York (NY) 2020. This software uses Hammett and Taft methods together with ionization and tautomerization prediction tools providing a reliable pK_a prediction. The following citations have been included in the paper (a) Greenwood, J. R.; Calkins, D.; Sullivan, A. P.; Shelley, J. C, Towards the comprehensive, rapid, and accurate prediction of the favorable tautomeric states of drug-like molecules in aqueous solution. J. Comp. Aided Mol. Des 2010, 24, 591-604. (b) Shelley, J. C.; Cholleti, A.; Frye, L.; Greenwood, J. R.; Timlin, M. R.; Uchimaya, M. Epik: a software program for pK_a prediction and protonation state generation for drug-like molecules. J. Comp. Aided Mol. Des. 2007, 21, 681-691. The calculation details have been included in the Supp. Info.

2. In the experimental section about experiments with glucose oxidase (page 9), DMSO appears unexpectedly in the reaction mixture (for protein stabilisation). The authors do not consider the effect of DMSO on thiol oxidation, although it is widely known and cited in the introduction (ref. 13).

The total amount of DMSO in the DCLs is 2.5% (v/v), and it is added to ensure the solubility of all library members at equilibrium. Figure 6 shows the quantitative measure of the rate acceleration in the homodimerization reaction of thiol 3. The accelerating role of DMSO was discarded by running control experiments containing up to 20% (v/v) DMSO, which is much larger than the amount of this co-solvent used in our standard conditions.

3. Paper is overloaded by unnecessary curiosities, such as: mentioning glucose oxidase applications (refs. 33-35 and refs. 44-46) and mentioning the biological role of spermine (which is doubled by the way, ref. 31 and refs. 38-40).

Thanks for the referee's suggestion. We have removed some of the above mentioned references and leave the most relevant ones.

4. The yellow circles symbolising the -SH groups in Figures 3 and 6 are not very visible. The contrast of yellow blocks should be increased.

The -SH symbol has been changed from empty to full yellow color symbol.

5. Figure 2 is unclear. The bonds and electric charges symbols are almost invisible/overlay. This should be corrected.

This figure has been corrected.

6. There is no information in Supplementary Figures 2-20 about, if is the isolated product chromatogram or extracted ion chromatogram and about formula of observed mass ions. The structural formula, exact mass, and calculated/found ion adduct is obligatory here.

The Supplementary Information regarding those figures has been completed with the extracted ion chromatogram, the formula of the observed mass ions, the exact mass on the synthesized compounds (3)₂, (4)₂, (5) and the calculated/found ion adducts.

7. In Supplementary Information, the authors claimed that "All chemicals and solvents were used from commercial sources such as [...]" (page 12), next they said, that components 1 and 2 were synthesised according to previously reported method (page 14). There is no characterisation of them (NMR or MS).

The characterization of compounds 1 and 2 has been completed in the Supplementary Information and the text corrected. New compounds have been synthesized and their characterization has been added to the Supplementary Information.

8. In Supplementary Information, there is a compound (6)₂ containing the unlabeled component 6 (page 14-15). I think it should be (3)₂ according to its name 3,3'-disulfanediyldibenzoic acid (page 15).

This was a mistake and we have corrected it, thank you.

Reviewer #2.

General suggestions:

1) As I said before, the results will be highly appealing for a reduced scientific community, working in the DCC field. It shows an apparently practical methodology for its use in specific applications. However, the work would be even more attractive for the general scientific community if the authors can show the generalized application of the method to the formation and exchange of disulfide bonds in, for instance, the correct disulfide bond in synthetic peptides containing Cys, or in the folding of a small protein.

Thanks for the referee suggestion. We have studied selenocystine properties for protein folding of scrambled RNase, an oxidized form of RNase A with random distribution of four disulfide bonds, and compared to the standard glutathione redox buffer (GSSG/GSH) pair (Figure 7). The RNase A activity was monitored following the hydrolysis of the cCMP (cytidine 2':3'-cyclic monophosphate monosodium salt). The refolding rate increased 5 times by replacing GSSG with Sec_{ox}. Unlike GSSG, Sec_{ox} is highly active even at acidic pH (5.4), expanding the range of conditions for protein refolding. These results have been included in the manuscript and Supplementary Information.

2) Alternative to the previous point, if the methodology can be used in systems where the slow disulfide oxidation and exchange are impractical but becomes fast and efficient with the selenocystine, the relevance of the proposed method would be higher. Despite I understand that these two points (1 and 2) are not essential for acceptance in Nature Communications, I believe they would make the paper more attractive, thus improving the potential impact of the work.

Thanks again for this suggestion. We have tested selenocystine in a particularly slow DCL made of 4 BBs, three of them with thiols $pK_a \geq 9$. Due to solubility issues a 20% (v/v) DMSO was added and the reaction mixture was diluted. In the absence of 5% (mol) Sec_{ox} (with respect to total thiol concentration), the system reached equilibrium after 192 h, whereas in the presence of selenocystine the system equilibrated after 60 h, i.e. three times faster. This experiment has been mentioned in the manuscript and the chromatograms have been included in the Supplementary Information.

3) The authors argued that they were inspired by a natural enzyme, but the similarity is rather low. The enzyme does not use diselenide bonds. They should downgrade that bio-inspiration or clearly underline the differences between the enzyme and their proposal.

We have clarified the points of the mechanism in common with the TrxR. The operating principle of this chemistry is that selenenylsulfide intermediates are more reactive towards thiolate nucleophilic additions, thus accelerating the exchange between the different species. Selenenylsulfide intermediates were detected in the DCL and the LC-MS evidence has been included in the Supplementary Information (Supplementary Figures 31-34).

4) Another fact that slightly reduces the innovative aspects of the work is the use of known dynamic combinatorial mixtures against known templates. Have the authors tried to apply it to a new non-reported system? Do the authors envision their general application?

Spermine was used to eliminate a possible conflict between the templating effect and selenocystine catalysis in a previously well studied 2 building blocks DCL. In this revised version of the manuscript we have applied Sec_{ox} in a different system such as our 4

building blocks DCL using spermine, spermidine and NADPH. Moreover, the enzyme glucose oxidase has not been previously used as protein template in a DCL.

We find that the presence of selenocystine, a commercially available reagent, produces a positive effect promoting the thiol/disulfide exchange and that it can be used as reagent in biological redox events such as the formation of the right disulfide bonds in the folding of small proteins.

5) Regarding the proposed mechanism through the formation of Se-S intermediate bond, do the authors have any experimental evidence of such species? A simple ESI-MS identification of this intermediate would be good enough, considering that intermediate is expected to exist in very low concentration.

We have included in the Supplementary Information the experimental evidence (ESI-MS identification) of S-Se intermediates that were detected in LC-MS. More specifically the species detected were 4-Sec, 3-Sec, 3-1-Sec and 3-2-Sec (Supplementary Figures 31-34).

Technical issues:

1) The authors said that the presence of different amounts of selenocystine does not change the DCL composition. By a close inspection of figure 5, this is unclear to me, since the relative heights of the (3)(2)(4) or the (2)4 peaks are slightly different in some experiments. Sometimes this is difficult to assess by naked eye, and the superposition of the spectra or the comparison of the integrated relative areas are more reliable (see, for instance, Figure 6).

The referee is right and relative peak areas (RPAs) could have been included in this experiment. Now, we have added the RPAs of the HPLC chromatograms in Figure 5b. The Tables with the percentages of all chromatograms where RPAs have been calculated are included as Supplementary Tables. In the DCLs included in Figure 5, this percentages are constant and in agreement with the experiments in absence of selenocystine confirming that the percentage of selenocystine does not alter the DCL composition.

2) In the GOx section, the amplified molecule could be a ligand for any type of hydrophobic pocket. Have the authors tested if this is the case using a dummy protein such as BSA or HSA. These could be rather conflictive since these proteins have several disulfide bonds. However, if the method also works in these cases, its impact would be higher.

Thanks for the referee's suggestion. Unspecific binding in a DCL can be discarded either by using BSA as a "dummy" protein or by the addition of a known inhibitor of the protein target to the DCL. We have performed both tests in our DCL. First the GOx was replaced by the BSA. The BSA-templated DCL did not amplify (4)₂, but it modified the DCL distribution as it was previously reported for thiol-disulfide exchange (see Supplementary

Figure 49 and Supplementary Methods).¹ Therefore, we performed the DCL in the presence of a large excess of DTNB (5,5'-dithiobis-2-nitrobenzoic acid), an inhibitor of GOx, which prevented the amplification of any other ligands except DTNB (see Supplementary Figure 50), indicating that an inhibited GOx cannot influence the DCL equilibrium composition (4)₂. We have incorporated the chromatograms in the Supplementary Information and the corresponding text in the manuscript.

Furthermore, in this revised version the amplified compound (4)₂ has been tested in enzymatic assays showing a non-competitive inhibitor behavior.

3) Related to that, the fluorescence titration of the enzyme with the ligand and glucose is not very informative. First of all, the authors should mention (main article or supporting information) which method or equation they used for the fitting. Moreover, the observed binding could be unspecific in this case. Have the authors checked the GOx activity with/without the amplified ligand?

Thanks for the referee's suggestion. We have realized that there was a mistake in the fluorescence binding data. The binding constant (K_d) was not mM but it was 110 μ M, and we have corrected it (data in Supplementary Information). The equations used for the fitting have been included in the Supplementary Methods.

The GOx activity with and without the ligand was evaluated by the horseradish-peroxidase system using o-dianisidine as a chromogen. Compound (4)₂ obtained a K_i value of $1.7 \pm 0.2 \mu$ M, showing a non-competitive inhibitor binding mode (Figure 10).

4) During the evaluation of the manuscript, I was quite surprised about the use of selenocystine in aqueous media, because this molecule is extremely insoluble in water. After the reading of the supporting information, I realized that the selenocystine was included in the reaction adding DMSO as a cosolvent. On the other hand, it is known that small amount of DMSO can also increase the disulfide exchange (as the authors cited in the introduction). Can the authors be sure that the increase in the rate of reaching the equilibrium is exclusively due to the presence of the diselenide? Do the corresponding reactions without added thiol, with Cys or with cystine also contain the same amount of DMSO as with selenocystine? In case of needed, the corresponding control experiments might be performed to confirm that the observed effects are due to the diselenide catalyst.

Due to solubility issues selenocystine was added in DMSO with 4% (v/v) 1M NaOH. We have included this information in Methods and Supplementary Information.

To ensure that the reaction was fully soluble 2.5% (v/v) DMSO was added in all the experiments (absence/presence selenocystine), so all our control (blank) experiments already got 2.5% (v/v) DMSO. On the revised version of the manuscript, in Figure 6 we have presented the quantitative measure of the rate acceleration on the homodimerization

¹ Danieli, B.; Giardini, A.; Lesma, G.; Passarella, D.; Peretto B.; Sacchetti, A.; Silvani, A.; Pratesi, G.; Zunino, F. Thiocolchicine-Podophyllotoxin Conjugates: Dynamic Libraries Based on Disulfide Exchange Reaction. *J. Org. Chem.* **2006**, 71, 2848-2853

reaction of compound 3 to (3)₂ in the presence of selenocystine and DMSO. The accelerating role of DMSO was discarded by running control experiments containing up to 20% (v/v) DMSO, which is much larger than the amount of this co-solvent used in our standard conditions (2.5% (v/v) DMSO).

Other minor issues:

1) All the amino acids shown in Figure 3 have D-stereochemistry. I guess this is an error, please amend that. The same issue states for the structures in Figure 1.

Thank you. It has been corrected.

2) For every mention to % of a reagent, especially in the methods part, it should be clearly stated if this is volume, weight or molar percent, and with respect to what.

Thank you. It has been included.

3) Some of the disulfides formed in the libraries (i.e. (4)₂) are expected to be poorly soluble in buffered water. The authors must clarify if they checked that possibility and how. When quantifying species in a mixture, solubility problems must be carefully considered.

The DCL is run in the presence of 2.5% (v/v) DMSO due to solubility issues. Under these conditions (basic pH and 2.5% (v/v) DMSO) the DCL is fully soluble. Precipitation in a DCL would have led to a kinetic trap making impossible the thermodynamic equilibration. Compound (4)₂ is soluble up to 2mM concentration.

Reviewer #3.

1. The main problem of the MS is that the authors only describe the rate enhancement by comparing if the reactions have reached equilibrium after 24 h or 96 h, as characterised by HPLC. It is shown for different ratios of the catalyst, but it doesn't provide any quantitative measure of the rate acceleration. And since this is the main point of the study, I find the manuscript too premature for publication in Nature Communication. The authors should provide quantitative characterisation of the rate constants, starting with a simple homodimerisation of thiols, as it is done in ref. 15.

To quantify the catalytic efficiency of Sec_{ox} in thiol-disulfide exchange, kinetic studies were performed and included in the manuscript. Figure 6b shows the time course for the (3)₂ homodimerization during the first 13 hours and the derived kinetic parameters (Figure 6c). In the presence of 5% (mol) Sec_{ox} the homodimerization reaction is completed in 24 h. The addition of 5% Sec_{ox} ($K_{obs}: 0.259 \pm 0.009 M^{-1} \cdot s^{-1}$) led to a 18-fold acceleration over the uncatalyzed reaction ($K_{obs}: 0.014 \pm 0.001 M^{-1} \cdot s^{-1}$). The reaction rate obtained with 5% (mol) of selenocystine is superior to those observed with 20% (v/v)

DMSO (K_{obs} : $0.068 \pm 0.001 M^{-1}\cdot s^{-1}$) and 5 % (mol) recombinant human TrxR 2 (K_{obs} : $0.035 \pm 0.003 M^{-1}\cdot s^{-1}$). Of note, the kinetic rate observed with 5% (mol) selenoenzyme resulted lower than with 1% (mol) Sec_{ox} (K_{obs} : $0.091 \pm 0.002 M^{-1}\cdot s^{-1}$).

2. In addition, the authors should also refer more precisely to prior work. When describing the building blocks, they should refer to prior work by Sanders and Otto that have used the same structure. There are also more studies on the spermine templated disulphide DCC than ref. 41 (See Journal of Systems Chemistry 2013, 4, 2 and Chem. Commun., 2009, 3708–3710 for both building blocks and spermine). On the other hand, the authors can leave out some of the many references to glucose which seem unnecessary.

The references including prior work with the building blocks have been included in the manuscript, as well as the references suggested by the referee. Besides, the references regarding the enzyme have been revised.

3. For the glucose templated DCC where 10% mol GOx is used, I am wondering how it can result in what seems to be approx. 20% mol of the binder (4)₂, especially since it has a binding constant of 1.9 mM to GOx while the building blocks and glucose are present in micromolar concentrations. Please explain.

The relative peak area (RPA) reports the oligomers distribution in templated or blank DCLs independently. However, the normalized change of RPA allows comparison between templated and non-templated DCLs. In our GOx example the percentage found for compound (4)₂ is 39%. This high percentage is because in one hand there are few oligomers detected in the templated DCL ((4)₂ gets a high RPA value) and on the other hand, compound (4)₂ was barely present in the non-templated DCL (low RPA value). The relation between these values in the normalized change of RPA equation is included in Supplementary Discussion. This percentage does not correlate to the amount of protein added to the library and it is relative to the other species in solution.

4. In the experimental section in the MS, please provide the molar amount of the chemicals in addition to the concentration and volume.

Thanks for the referee's suggestion. We have introduced the molar amounts of all the chemicals used in addition to the concentration and volume.

Reviewer #1 (Remarks to the Author):

I thank the authors for their careful compliance with the referee's comments. However, I remain unconvinced about the novelty aspect of this work and in my opinion the article concerns too narrow peace of chemical sciences for publication in Nature Communications. In my opinion, authors should consider splitting this voluminous material into two articles; the first might be devoted entirely to the selenocysteine catalysis in combinatorial chemistry for the DCC auditory; the second for application of selenocysteine SS catalysis in enzymatic / biological research for a dedicated audience in the specialized journal.

Reviewer #2 (Remarks to the Author):

The authors have considered all the concerns form the reviewers, answering to key questions raised after reading the first version of the manuscript. The quality of the manuscript has been highly improved after the corrections and additions done by the authors. Besides, they have carried out additional experimental work and they have further analyzed the data under a broader perspective. The kinetic analyses and the inhibition assays are specially noteworthy. Also the use of the selenocystine buffer for the folding of an active protein increases the generality of the procedure, not only useful for DCC researchers. This will increase the potential impact of the paper. I am fully supporting the acceptance of the manuscript in Nature Communications in the current format, just needing some minor polish that can be done at the editorial management of the materials.

I just wanted to add a personal note on the very professional attitude of the authors of this manuscript. They took the honest concerns from the referees as a constructive contribution rather than a mere criticism. The final outcome is a highly improved, more general, potentially more impactful piece of work with enough merit to be published in a top-rated scientific journal. The two young contributors in this manuscript should be proud of the work done here and their behavior during the revision process.

Reviewer #3 (Remarks to the Author):

I do believe the authors have addressed the criticism of the manuscript satisfactorily, in particular by including the kinetic experiments. I have no further reservations to the acceptance of the manuscript.

We would like to thank you for reviewing our manuscript “Biomimetic selenocystine based dynamic combinatorial chemistry for thiol-disulfide exchange”. We sincerely appreciate all the valuable comments which have help us to improve the quality of the manuscript.

Response to the reviewer 1:

Reviewer #1 (Remarks to the Author):

I thank the authors for their careful compliance with the referee's comments. However, I remain unconvinced about the novelty aspect of this work and in my opinion the article concerns too narrow peace of chemical sciences for publication in Nature Communications. In my opinion, authors should consider splitting this voluminous material into two articles; the first might be devoted entirely to the selenocysteine catalysis in combinatorial chemistry for the DCC auditory; the second for application of selenocysteine SS catalysis in enzymatic / biological research for a dedicated audience in the specialized journal.

Although it could be interesting to divide this work into two independent manuscripts we believe that the proof of concept and application are better described in their current form. Besides, the other two referees who revised the manuscript have considered of interest to include the biological application in it. We would therefore prefer to keep it in the current format.